# Exploring Pyrrolo-Fused Heterocycles as Promising Anticancer Agents: An Integrated Synthetic, Biological, and Computational Approach

**DOI:** 10.3390/ph16060865

**Published:** 2023-06-11

**Authors:** Roxana-Maria Amărandi, Maria-Cristina Al-Matarneh, Lăcrămioara Popovici, Catalina Ionica Ciobanu, Andrei Neamțu, Ionel I. Mangalagiu, Ramona Danac

**Affiliations:** 1TRANSCEND Research Center, Regional Institute of Oncology Iasi, 2-4 General Henri Mathias Berthelot Street, 700483 Iasi, Romania; rpomohaci@iroiasi.ro (R.-M.A.); neamtuandrei@gmail.com (A.N.); 2“Petru Poni” Institute of Macromolecular Chemistry of Romanian Academy, 41A Grigore Ghica Voda Alley, 700487 Iasi, Romania; 3Faculty of Chemistry, Alexandru Ioan Cuza University of Iasi, 11 Carol I, 700506 Iasi, Romania; tiru.lacramioara@yahoo.com (L.P.); ionelm@uaic.ro (I.I.M.); 4Institute of Interdisciplinary Research-CERNESIM Centre, Alexandru Ioan Cuza University of Iasi, 11 Carol I, 700506 Iasi, Romania; catalina.ciobanu@uaic.ro

**Keywords:** anticancer, cycloaddition, molecular docking, molecular dynamics, pyrroloisoquinoline, pyrroloquinoline, phenstatin analogs

## Abstract

Five new series of pyrrolo-fused heterocycles were designed through a scaffold hybridization strategy as analogs of the well-known microtubule inhibitor phenstatin. Compounds were synthesized using the 1,3-dipolar cycloaddition of cycloimmonium N-ylides to ethyl propiolate as a key step. Selected compounds were then evaluated for anticancer activity and ability to inhibit tubulin polymerization in vitro. Notably, pyrrolo[1,2-*a*]quinoline **10a** was active on most tested cell lines, performing better than control phenstatin in several cases, most notably on renal cancer cell line A498 (GI_50_ 27 nM), while inhibiting tubulin polymerization in vitro. In addition, this compound was predicted to have a promising ADMET profile. The molecular details of the interaction between compound **10a** and tubulin were investigated through in silico docking experiments, followed by molecular dynamics simulations and configurational entropy calculations. Of note, we found that some of the initially predicted interactions from docking experiments were not stable during molecular dynamics simulations, but that configurational entropy loss was similar in all three cases. Our results suggest that for compound **10a**, docking experiments alone are not sufficient for the adequate description of interaction details in terms of target binding, which makes subsequent scaffold optimization more difficult and ultimately hinders drug design. Taken together, these results could help shape novel potent antiproliferative compounds with pyrrolo-fused heterocyclic cores, especially from an in silico methodological perspective.

## 1. Introduction

Pyrrole and its hetero-fused derivatives have sparked great interest in the field of medicinal chemistry as valuable scaffolds for generating new drugs with biological activity, including anticancer, antimicrobial, or antiviral compounds [1]. Semisynthetic pyrroloquinolines are currently used as first-line agents in cancer therapy, including camptothecin analogs irinotecan and topotecan, with many others currently being investigated as potent antiproliferative drugs [2]. Pyrrolo[2,1-*a*]isoquinoline is a common motif in various bioactive alkaloids such as lamellarin D, which is known for its topoisomerase I inhibitory properties [3], or the antiproliferative alkaloid crispine A [4]. Pyrrolopyrimidine-based derivatives such as pemetrexed, immucillin H, or variolin B are also used as anticancer drugs. In addition, several other pyrrolo[1,2-*a*]pyrazine alkaloids, such as dibromofakelin and longamide B, exhibit multiple biological activities, including cytotoxicity [5]. Moreover, our group has reported pyrrole-fused heterocyclic derivatives with an indolizine, pyrrolopyridazine, pyrrolo(iso)quinoline, and pyrrolophenathroline core, with part of them showing considerable anticancer properties [6,7,8,9,10,11,12,13,14].

Of note, many pyrrole-fused heterocyclic ligands exhibit antitumor activity by binding to the colchicine site of tubulin, which is particularly known for its ability to accommodate a wide variety of structurally diverse ligands [15]. Compounds that target the colchicine site mainly exert their biological effects by inhibiting the formation of the cell mitotic spindle, causing arrest in the G2/M phase of the cell cycle, eventually triggering apoptotic cell death [16]. Since the identification of the colchicine binding site in 2004 through the 3.5 Å resolution crystal structure of the α,β-tubulin heterodimer and N-deacetyl-N-(2-mercaptoacetyl) colchicine (DAMA-colchicine) [17], several other colchicine site-binding agents have been co-crystallized with tubulin [18], paving the way to structure-based drug design campaigns aimed at identifying novel antitumor agents with various scaffolds [19,20,21]. At the same time, the abundance of published protein structures with diverse inhibitors allows for the direct comparison of co-crystallized and predicted conformations for a multitude of biologically active agents. However, results do not always agree, often leading to very different directions in structural optimization strategies [22,23]. In the case of tubulin, the recently published crystal structures of several heterocyclic-fused pyrimidines revealed 180° flipped binding poses when compared to previous molecular docking predictions [24]. As such, it is increasingly evident that simple molecular docking approaches for ligand binding pose prediction by using X-ray structures of tubulin co-crystallized with structurally different molecular scaffolds can be challenging and have limited reliability. Although useful for medicinal chemistry, structural insights obtained from molecular docking studies alone regarding the interaction between pyrrole/pyrrole-fused heterocyclic derivatives and tubulin [6,25,26] can ultimately confuse novel design and development efforts in this class of tubulin inhibitors due to the possibility of questionable interpretation [27].

Molecular dynamics (MD) can be a useful tool for estimating the stability of ligand-receptor conformations predicted by molecular docking and can be used as a utensil for docking pose validation [28,29]. As such, when the conformation of a ligand after an MD simulation deviates by more than a specific RMSD value from the initial docking prediction, the docking pose can be regarded as unstable even though it has a high docking score [27,30]. Since MD has been described as useful for assessing the stability of several tubulin colchicine site binders, including combretastatin analogs [31] and benzimidazoles [32], as well as for investigating DAMA-colchicine binding to different tubulin isotypes [33], using such a strategy to validate molecular docking poses in the case of pyrrole-fused derivatives could be useful in the future development of more potent tubulin inhibitors containing this scaffold.

This study presents the development, synthesis, and assessment of the anticancer potential of newly designed pyrrolo-fused heterocycles, which include quinoline, isoquinoline, benzo[*f*]quinoline, pyrazine, and pyrimidine. The rationale behind compound design was based on the observation that by substituting the 3-hydroxy-4-methoxyphenyl ring of phenstatin (a tubulin polymerization inhibitor) with pyrrolo-fused heterocycles such as indolizine, pyrrolopyridazine, or pyrroloquinoline (Figure 1), compounds with excellent anticancer properties can be obtained [6,11,14,34,35].

The tubulin polymerization inhibition mechanism can be partly responsible for the anticancer activity of these compounds [6,11,34,35]. The best compound in the series, pyrrolo[1,2-*a*]quinoline **10a**, was further chosen for a series of in silico investigations, including global and local docking, molecular dynamics simulations, and configurational entropy calculations. This thorough in silico evaluation was conducted in order to describe the interaction of this compound with the colchicine binding site in detail, as well as investigate the stability of identified ligand-protein interactions through molecular docking experiments.

In the above context, the herein study was designed in order to identify novel anticancer tubulin-targeting agents that bind to the colchicine site, as well as describe the molecular details of hit binding through extensive in silico investigations. The broader goal of our investigation is to deepen the molecular understanding of biological activity and help shape future rational drug design campaigns aimed at the colchicine site of tubulin.

## 2. Results

### 2.1. Chemistry

The strategy to build the five series of target phenstatin analogs (Figure 1) consisted of two main synthetic steps, starting from the desired heterocycle to be fused to the pyrrole ring.

Four derivatives were synthesized in each series of compounds, differing by the substituent at the phenyl ring. First, monoquaternary salts **3**, **6**, **9**, **12**, and **15** were prepared by the direct reaction of pyrazine **1**, 4-(4-chlorophenyl)-pyrimidine **5**, quinoline **8**, benzo[*f*]quinoline **11**, or isoquinoline **14**, respectively, with 2-bromo-acetophenones **2** in acetone, at room temperature (r.t.) (Figure 1, Figure 2, Figure 3, Figure 4 and Figure 5). The reaction of pyrimidine in similar conditions did not produce the desired salts, and that is why we used the substituted pyrimidine derivative **5** (previously synthesized by our group [36]) as the starting material for the target compounds **7a**–**d**.

We employed the 1,3-dipolar cycloaddition method to synthesize pyrrolopyrazine/pyrrolopyrimidine/indolizine rings. In this method, we generated heterocyclic ylides in situ from the monoquaternary salts **3**, **6**, **9**, **12**, and **15** and reacted them with ethyl propiolate in a basic medium. The reaction schemes for each series are illustrated in Figure 1, Figure 2, Figure 3, Figure 4 and Figure 5. As expected, cycloadditions occurred with the selective formation of the regioisomers **4**, **7**, **10**, and **13**, which is in agreement with the electronic effects within ethyl propiolate and ylide species.

All 4-bromo-substituted intermediate salts (**3d** [37], **6d** [36], **9d** [6], **12d** [38], and **15d** [6]) and 4-bromobenzoyl-cycloadducts (**7d** [36], **10d** [39], **13d** [40], and **16d** [39]) have been previously reported in the literature. However, the compounds were synthesized and subsequently tested for their anticancer activity in order to provide a more detailed understanding of the structure-activity relationships (SAR).

The fused target compounds were obtained in moderate yields ranging from 30% to 80%. The structures of all intermediate and final compounds were fully confirmed by spectral and physicochemical data (Appendix A). The structures of the cycloadducts presented in Figure 1, Figure 2, Figure 3, Figure 4 and Figure 5 were confirmed by NMR and IR spectra. These spectra are consistent with those of similar compounds previously reported in the literature [6,9], where clear structural evidence was emphasized, including X-ray diffraction.

### 2.2. Biological Activity

#### 2.2.1. Anticancer Activity

All forty synthesized compounds (monoquaternary salts and cycloadducts) were electronically submitted to the National Cancer Institute (NCI) platform, and twenty-five compounds (**4b**–**d**, **7b**–**c**, **9a**, **9d**, **10a**–**d**, **12a**–**d**, **13a**–**d**, **15a**, **15d**, and **16a**–**d**) were selected for single dose (10^−5^ M) screening against a panel of 60 human tumor cell lines, representing leukemia, melanoma, and cancers of the lung, colon, central nervous system, ovary, kidney, prostate, and breast [41]. Selected representative results for best hits and phenstatin are summarized in Table 1, all the other tested compounds being inactive against the NCI cancer cells at 10^−5^ M (results provided by NCI for all tested compounds can be found in Appendix A).

Compound **10a**, which contains a pyrrolo[1,2-*a*]quinoline scaffold, showed excellent growth inhibitory properties against almost all tested cell lines, with an average GP (growth percent—growth of treated cultures relative to untreated cultures) for all tested cell lines of 25.7% (74.3% GP inhibition). This compound exhibited the best inhibitory activity on the growth of leukemia cell lines, with an average GP inhibition of 87.33%, prostate cancer lines (78.0% GP inhibition), and breast cancer lines (76.4% GP inhibition). Compound **10a** also had a cytotoxic effect on several cell lines, most notably on melanoma MDA-MB-435 cells, where 54.59% of cultured cells were killed at a 10^−5^ M concentration. Even if phenstatin was superior in terms of average GP inhibition, compound **10a** showed similar or even better inhibitory properties against several cell lines. Interestingly, the other three compounds from this series, **10b**–**10d**, did not exert any relevant inhibitory activity, suggesting that the 3,4,5-trimethoxyphenyl group plays a very important role in the observed anticancer activity in this series of compounds.

From the other series of synthesized cycloadducts, only pyrrolo[2,1-*a*]isoquinoline **16b** exhibited specific cytotoxic activity against the SNB-75 CNS cancer cell line, while compound **13b**, a benzo[*f*]pyrrolo[1,2-*a*]quinoline, was moderately active in inhibiting the growth of SR leukemia cells (71% GP inhibition).

Among the intermediate salts, benzo[*f*]quinolin-4-ium bromides **12** showed moderate inhibitory properties against the growth of colon HCT-116 cells and breast MDA-MB-468 cells, with growth inhibition values of approximately 50%.

Compounds that fulfilled predetermined threshold inhibition criteria were then chosen by the NCI for the second testing step to determine GI_50_, TGI, and LC_50_ parameters. Showing the most significant growth inhibition, compound **10a** was the only one selected for evaluation against the sixty tumor cell lines in five-dose assays [41,42,43]. The most representative results from the NCI-60 five-dose screen are shown in Table 2 (full results can be found in Appendix A).

Notably, compound **10a** displayed almost all GI_50_ values in the nanomolar concentration range. This pyrrolo[1,2-*a*]quinoline derivative outperformed control phenstatin in several cases, including renal cancer cell line A498, colon cancer cell line COLO 205, and breast cancer cell line T-47D. Furthermore, compound **10a** demonstrated good performance against various cancer cell lines (including melanoma MDA-MB-435, renal cancer A498, ovarian cancer OVCAR-3, breast cancer MDA-MB-468, and non-small cell lung cancer NCI-H522 cell lines), exhibiting complete growth inhibition at submicromolar concentrations (Table 2). However, LC_50_ values (the molar concentration of tested compound causing 50% death of tumor cells) for compound **10a** and phenstatin, respectively, were found to be >100 μM against all tested lines.

NCI’s in silico platform, PRISM, enables investigators to compare their active molecules with other molecules that have been previously tested through NCI from both a biological and chemical perspective [44]; this evaluation is performed with two components: COMPARE and PILOT. Interestingly, when analyzing the most active compound **10a** with PRISM, we found a best-fitting profile with N-(4-bromophenyl)-4-(2-cyclopropyloxazol-5-yl)benzenesulfonamide [45], which is a compound from a series of reported anticancer derivatives and is also able to inhibit tubulin polymerization, as we expect for our compounds.

#### 2.2.2. In Vitro Tubulin Polymerization Inhibition

To confirm that the observed anticancer activity of compound **10a** is conferred by a microtubule-targeting mechanism, we evaluated its effect on the assembly of tubulin, using paclitaxel (a tubulin stabilizer) and phenstatin (a tubulin polymerization inhibitor) as controls. As expected and shown in Figure 2, paclitaxel was found to stimulate tubulin polymerization, while phenstatin and compound **10a** inhibited tubulin polymerization, although **10a** had a slightly weaker effect. The obtained data indicate that compound **10a** effectively inhibits tubulin polymerization in vitro, suggesting that the observed cytotoxicity of this compound is related to a microtubule-targeting mechanism.

### 2.3. Molecular Modeling

#### 2.3.1. Blind Docking

Since compound **10a** is similar to phenstatin in inhibiting tubulin polymerization in vitro, we hypothesize that it binds to the colchicine site of the α,β-tubulin heterodimer, the known site for phenstatin [46]. In addition, recent tubulin crystal structures co-crystalized with quinazolinone and tetrahydroisoquinoline anticancer agents reveal the preference of these heterocyclic compounds to the colchicine binding site [47,48] and support our previous hypothesis that quinoline derivatives could exert their anticancer activity by binding to the colchicine site of tubulin [6]. However, tubulin binding sites other than colchicine have been described, including the vinca, taxol, and peloruside/laulimalide sites [49], and other novel sites are being actively discovered and/or investigated for the rational design of tubulin modulators [50,51] or compounds targeting drug resistant cancer phenotypes [19,52]. Thus, we performed blind docking experiments for compound **10a**, in order to investigate its relative preference towards the colchicine binding site, while validating our docking protocol using colchicine and phenstatin (Appendix A).

The generated poses for compound **10a** had estimated binding energies between −8.2 and −2.0 kcal/mol, with an overall score distribution and RMSD from best-scoring conformation for all docked poses resembling both colchicine and phenstatin (Appendix A). Clustering with a 2.0 Å RMSD tolerance gave 127 unique cluster representatives for **10a**, 13 of which scored lower than −7 kcal/mol. Out of these, the lowest-scoring conformation and an additional 10 cluster representatives were positioned in the colchicine binding site. All colchicine cluster representatives with estimated binding energies lower than −7 kcal/mol were positioned in the colchicine binding site of tubulin, as expected. RMSD between co-crystallized colchicine and lowest-scoring cluster representative from the blind docking was 1.082 Å, which is lower than the 2.0 Å cutoff generally used as a criterion for correct bound structure prediction [53], suggesting that the used docking protocol is suitable for the studied system. None of the phenstatin conformations scored less than −7 kcal/mol, but the four lowest-scoring cluster representatives were also positioned in the colchicine binding site. These results suggest that compound **10a** prefers the colchicine binding site of tubulin and is likely able to inhibit tubulin polymerization by binding to the colchicine binding site of tubulin while having low affinity for other tubulin sites.

#### 2.3.2. Local Docking

As the blind docking experiments revealed that the most energetically favorable poses for compound **10a** are positioned in the colchicine binding site of tubulin, we performed local docking on this site in order to investigate the molecular nature of these preferential conformations. A clear improvement in the overall docking score distribution can be observed when compared to blind docking due to the decrease in conformational search space, as expected (Appendix A). Interestingly, three low scoring clusters of comparable energies (−8.95 kcal/mol, −8.77 kcal/mol, and −8.69 kcal/mol—conformations I, II, and III, respectively) were revealed through local docking and were accommodated in the colchicine binding site in three geometrically different modes (BM I, BM II, and BM III). Of note, BM III greatly overlapped with the lowest-scoring cluster representative from blind docking for compound **10a** (RMSD 1.034 Å). These three lowest-scoring cluster representatives were further chosen for analysis, as there have been cases of false positives in top-scoring poses from docking experiments, and subsequent analysis of multiple low-scoring docking poses is strongly recommended to correctly identify the most likely ligand conformation [54,55]. The RMSD between these modes was 7.51 Å and 6.408 Å (using BM I as a reference, as it is the lowest-scoring cluster), and the distances between their centers of mass were 3.197 Å (BM I/BM II), 3.294 Å (BM I/BM III), and 5.145 Å (BM II/BM III), indicating that the three conformations greatly differ from one another. The three best-scoring docking solutions from local docking also varied in terms of key amino acids involved in ligand stabilization at the binding site (Figure 3). For docking protocol validation, colchicine was re-docked in the same manner as **10a**. RMSD between co-crystallized colchicine and the lowest-scoring cluster representative after colchicine re-docking was 1.110 Å.

In the case of BM I, the pyrrolo[1,2-*a*]quinoline ring is buried in the hydrophobic pocket lined by βCys241, βLeu242, βLeu252, βLeu255, βMet259, βVal315, and βAla316, while the trimethoxy-substituted ring is positioned towards the α subunit, in a manner similar to what we have previously seen for moderately active isoquinoline cycloadducts [6], yet very dissimilar to colchicine (Appendix A). This conformation is further stabilized through hydrogen bonding with the sidechain of βLys254, an amino acid that has been previously identified through molecular docking experiments as an interaction partner for other tubulin polymerization inhibitors with anticancer activity [56,57,58].

However, to our knowledge, none of the co-crystallized tubulin inhibitors that bind to the colchicine site have been shown to interact with this residue, although it has been highlighted that this amino acid is likely involved in microtubule assembly through interaction with the γ-phosphate group of the N-site GTP [59]. Furthermore, alanine scan mutations have shown that D251A-R253A-K254A is lethal in yeast [60]. In addition, a recent molecular dynamics study focused on the dynamics of the βT7 loop concluded that an interaction between the backbone N atom of βLys254 and the side chain O atom of βAsp251 is essential for the structural rearrangement of the βT7 loop [61], which has been shown to play an important role in colchicine binding [62] and prevents the ‘curved-to-straight’ structural transition of tubulin from its free form, a process that is necessary for microtubule formation [63]. Therefore, βLys254 could be a likely interaction partner for compound **10a**.

BM II was stabilized in the colchicine binding site exclusively through hydrophobic interactions, having an orientation similar to what we have previously predicted for active isoquinoline derivatives [6]. However, no hydrogen bond interaction with βCys241, the presumed anchor point for the trimethoxy-substituted ring, was observed. Nevertheless, the trimethoxy-substituted ring roughly occupies the same hydrophobic pocket as the trimethoxy-substituted moiety of colchicine (Appendix A). Since some colchicine site binders, including 2-aroylindoles, have been shown to be stabilized in the colchicine site exclusively through hydrophobic interactions and water-mediated polar interactions that do not include βCys241 [64,65], such a conformation could also be probable for compound **10a**. In fact, this conformation occupies hydrophobic center II according to a previously described structure-based colchicine binding-site inhibitor model [21] and greatly overlaps with 2-aroylindole tubulin polymerization inhibitor D64131 [65], with an additional hydrophobic extension towards the α subunit (Appendix A).

In BM III, the trimethoxy-substituted ring was oriented towards the α subunit and engaged in hydrogen bonding with many amino acid sidechains known or thought to be involved in colchicine binding site inhibitor interaction, including αAsn101 [51], αThr179 [65], αSer178 [31], βLys352 [66], and βGln247 [67]. However, to our knowledge, no other colchicine site inhibitors possessing a trimethoxy-substituted ring have been co-crystallized in a similar conformation [18]. This conformation is also the most dissimilar from colchicine (Appendix A).

Importantly, all three lowest cluster representatives of **10a** had similar theoretical binding energies. While scoring functions take into account a wide range of contributions, including electrostatic interactions, van der Waals contacts, and desolvation effects [68], they are limited in reflecting the conformational changes induced by ligand binding, as well as the stability of identified amino acid interactions in time.

#### 2.3.3. Molecular Dynamics Simulations

To investigate the detailed dynamics and interaction stability of ligand-tubulin complexes, the tubulin heterodimer and the three best-scoring local docking poses were subjected to 10 ns MD simulations.

Overall, all three systems reached equilibrium, according to root mean square deviation (RMSD) analysis (Figure 4a), even though RMSD was higher for all investigated binding modes when compared to reference colchicine. In addition, BM I shifted from the initial frame more than BM II or BM III (Figure 4b), having a mean RMSD of 1.154 Å, while BM II and BM III had mean RMSD of 0.721 Å and 0.66 Å, respectively. All binding modes had a higher RMSD than colchicine (mean deviation 0.39 Å), which would be expected since the starting tubulin heterodimer structure is co-crystallized with colchicine. The ‘Lig fit Prot’ (Figure 4c) represents the RMSD of ligand-heavy atoms after first aligning the protein-ligand complex on the protein backbone. If the observed values differ greatly from the protein backbone RMSD, it is likely that the ligand diffuses away from the binding site [69]. In the case of BM I, this parameter shows drift during the simulation, particularly at the beginning of the trajectory, but is stabilized until the end of the simulation. This suggests that the initially found local docking conformation does not maintain close contact with surrounding amino acids and that another conformation is stabilized during the trajectory.

Indeed, for BM I, the 10 ns MD simulation revealed that the interaction with βLys254, which was the main anchor point for this docking conformation, was only transient, being maintained in less than 1% of the simulation time (Figure 5a). However, BM I maintained contacts in more than 25% of simulation time with αTyr224 (44.96%), βAsn249 (58%), βLys352 (53.35%), βAla250 (59.44%), βLeu255 (61.49%), βAla316 (37.91%), and the co-crystalized GTP molecule (39.51%). The ligand RMSD with regards to the initial docking conformation had the highest value of all simulated binding modes (3.543 Å). Superimposition between the docked solution and the final frame of the simulation revealed major differences in terms of amino acid contacts within the colchicine binding site (Appendix A). Of note, the carboxylate moiety flips after 3.57 ns and remains flipped throughout the simulation. The flexibility of this group is also evident in the ligand root mean square fluctuation—RMSF (Appendix A). Overall, the BM I-containing simulated system converged to the least similar conformation from the starting docking solution.

BM II maintained all the initially identified hydrophobic contacts, while slightly drifting from the initial docked conformation (RMSD 1.746 Å). In addition, a cation-pi interaction with βLys352 was observed in 27.32% of the simulation time. This binding mode was also the only one to engage in favorable interactions with βCys241, maintaining contact with this amino acid in 20.53% of simulation time (Figure 5b), mainly through the central methoxy moiety from its trimethoxy-substituted ring. The flexibility of this substituent in BM II is most evident from the abrupt increase or decrease in the main RMSD profile with respect to the first frame of the simulation (Figure 4b), as well as from the ligand root mean square fluctuation (Appendix A). Of note, RMSF of the same substituent in colchicine shows a similar behavior throughout the simulation (Appendix A).

In the case of BM III, the H-bonds with αAsn101, αSer178, and βGln247 identified from the docking solution were mostly not maintained during the MD simulation (Figure 5c), except for the initial frames (Appendix A). The H-bond with αThr179 was maintained in 15.03% of simulated time, either directly or mediated through a water molecule, while interaction with βLys352 was maintained throughout the entire simulation (Figure 5c). RMSD between the initial docked solution and the final frame of the MD simulation was 3.525 Å.

It is important to note that from all final frame MD simulation conformations, BM II was the only one that largely overlapped with co-crystallized colchicine binding site inhibitors with a trimethoxy-substituted ring [25,65,70], with most hydrophobic contacts being maintained throughout the simulation, even though the number of polar contacts was the smallest from all simulations (Appendix A). In addition, all simulated BMs for compound **10a** maintained various kinds of contacts with βLys352 in more than 25% of the simulated time, suggesting that this amino acid could be relevant for the binding of this compound in the colchicine site, regardless of the initially identified docking solution. The importance of this residue in **10a** binding could be further investigated through alanine scanning, as has been previously performed for the anticancer natural product pironetin [71].

Since receptor-ligand binding does not rely solely upon specific interactions between amino acids and particular chemical moieties but also on reduction of molecular flexibility upon binding [72], we further investigated the configurational entropy changes of all three binding modes upon interacting with the colchicine binding site of tubulin. This gave us an estimation of the entropic cost upon target binding. Since longer MD simulations are preferred in order to fully assess the stability of the investigated systems [30], and sufficient sampling should be reached to estimate configurational entropies [73], we extended our simulations to 25 ns, for which a good convergence of the configurational entropy profile was achieved. We found that the internal configurational entropy is reduced upon binding from 156.02 J/mol K to 99.52 J/mol K, 99.55 J/mol K, and 98.28 J/mol K, for BM I, BM II, and BM III, respectively (ΔS_conf_ = −56.50 J/mol K; −56.47 J/mol K; −57.74 J/mol K, respectively). Of note, BM II exhibited a markedly lower internal configurational entropy than BM I and BM III in the first quarter of the simulation but steadily increased to comparable values as the other two binding modes throughout the rest of the simulated time (Appendix A). Our calculated values are similar in magnitude to other receptor-ligand systems [73].

Overall, the molecular modeling outcomes enabled us to predict, from a structural perspective, the interactions between our novel compound and tubulin with atomic resolution. Our results suggest that all three binding modes to tubulin described here are characterized by both positive enthalpic and entropic contributions, highlighting that all three binding modes are equally probable in a biological context. The molecular mechanics energies combined with the Poisson–Boltzmann or generalized Born and surface area continuum solvation (MM/PBSA and MM/GBSA) methods [74] could also have been of particular use in assessing ligand binding affinities for each of the three poses, but they have not been explored in the current study.

#### 2.3.4. In Silico ADME and Toxicity Predictions

The result of the predicted parameters, including molecular properties, pharmacokinetics, drug-likeness, and medicinal chemistry are presented in Table 3.

Compound **10a** follows both Lipinski’s rule of five (without any violations) and Veber’s rule, having less than 10 rotatable bonds and a topological polar surface area smaller than 140 Å^2^. In addition, compound **10a** displays a moderately soluble behavior, with a Log S (ESOL) value of −6.00 and no PAINS or Brenk alerts in its structure. Derivative **10a** also scores well in terms of bioavailability and ease of synthetic accessibility (Table 3). This compound is predicted to have a high gastrointestinal absorption but is unable to permeate through the blood-brain barrier (BBB). Compound **10a** does not appear to be a P-glycoprotein (P-gp) substrate.

The chart generated from the Swiss ADME QSAR web tool, regarding the accessibility of compound **10a** to be orally bioavailable, is presented in Figure 6.

This radar involves six parameters, lipophilicity (LIPO), size, polarity (POLAR), insolubility (INSOLU), unsaturation (INSATU), and flexibility (FLEX), of the tested compound and is represented by a red line integrated into a pink area. Molecules that fall within the pink region of the radar are considered drug-like. Compound **10a** exhibited compliance to only four of the six rules, with violations to INSATU (ratio of hybridized sp3 atoms to the total number of C atoms) and LIPO (XLOGP3 between −0.7 and +5.0). Taken together, these predicted data show a promising ADME and drug-likeness profile for compound **10a**.

The predicted toxicity spectrum is represented by a list of activities with probabilities “to be active” (P_a_) and “to be inactive” (P_i_). The obtained results presented in Table 4 show predicted cytotoxicity (Pa > Pi and Pa > 0.3) against several cancer cell lines, including four of the tested NCI cell lines: T-47D, HT-29, DU-145, and MCF7 (Table 3). The fact that no normal human cell lines appeared on the list could be an indication for a good selectivity of compound **10a** against cancer cell lines.

## 3. Materials and Methods

### 3.1. Chemistry

All commercially available reagents and solvents employed were used without further purification. Melting points were recorded on an A. Krüss Optronic Melting Point Meter KSP1 and are uncorrected. Analytical thin-layer chromatography was performed with commercial silica gel plates 60 F254 (Merck Darmstadt, Germany) and visualized under UV light (λ_max_ = 254 or 365 nm). The NMR spectra were recorded on a Bruker Avance III 500 MHz spectrometer or a Bruker Avance 400 DRX (400 MHz) (Bruker, Vienna, Austria). Chemical shifts are reported in delta (δ) units, part per million (ppm), and coupling constants (*J*) in Hz. The following abbreviations are used to designate chemical shift multiplicities: s = singlet, d = doublet, t = triplet, q = quartet, m = multiplet, and bs = broad singlet. Infrared (IR) spectra were recorded as films on potassium bromide (KBr) pellets on an FTIR Prestige 8400 s spectrophotometer (Shimadzu, Kyoto, Japan) or a Jasco 660 FTIR spectrophotometer. Elemental analyses indicated by the symbols of the elements were within ± 0.4% of the theoretical values. HR-MS experiments were recorded on a HESI Orbitrap Exploris 120 Mass Spectrometer (Thermo Fisher, Walthan, MA, USA) in positive mode.

#### 3.1.1. General Procedure for Monoquaternary Salts **3**, **6**, **9**, **12**, and **15**

The corresponding heterocycle (pyrazine **1**, 4-(4-chlorophenyl)pyrimidine **5**, quinoline **8**, benzo[*f*]quinoline **11** or isoquinoline **14**) (1 mmol, 1 equiv.) was dissolved in 5–7 mL acetone. Then, the corresponding reactive 2-bromoacetophenone **2a**-**d** (1.1 mmol, 1.1 equiv.) was added and the resulting mixture was stirred overnight at room temperature (r.t.). The formed precipitate was filtered and washed with diethyl ether to obtain the desired product, which was used in the next reaction without any further purification.

#### 3.1.2. General Procedure for Compounds **4**, **7**, **10**, **13**, and **16**

The cycloimmonium salt (**3**, **6**, **9**, **12**, and **15**) (1 mmol, 1 equiv.) and ethyl propiolate (1.1 mmol, 1.1 equiv.) were added to dichloromethane (DCM). Then, a solution of triethylamine (TEA) (3 mmol, 3 equiv.) in DCM (3 mL) was added dropwise over 1 h (magnetic stirring), and the resulting mixture was stirred overnight at r.t. Methanol (10 mL) was added, and the resulting solid was collected by filtration and washed with 5 mL methanol. The product was then purified by crystallization from dichloromethane/methanol (1/1, *v*/*v*) and/or column chromatography using dichloromethane/methanol (99.5/0.5, *v*/*v*). Compounds **4**, **7**, **10**, **13**, and **16** were the only pure compounds obtained from the reaction mixture fractions. While spectral evidence for some decomposition products of the ylides was observed in the other fractions, no evidence was found for the presence of the other possible regioisomer cycloadducts.

#### 3.1.3. Spectral Data

##### 1-(2-oxo-2-(3,4,5-Trimethoxyphenyl)ethyl)pyrazin-1-ium Bromide **3a**

Brown solid; 50% yield; mp 160–162 °C; IR ν(cm^−1^): 3009, 2986, 1682, 1630, 1584, 1445, 1416, 1186, 1124; ^1^H NMR (400 MHz, DMSO-d_6_) δ_ppm_: 3.80 (s, 3H, OMe), 3.90 (s, 6H, 2 × OMe), 6.67 (s, 2H, H_7_), 7.38 (s, 2H, H_10_, H_14_), 9.19 (d, *J* = 4.0 Hz, 2H, H_3_, H_5_), 9.72 (d, *J* = 4.0 Hz, 2H, H_2_, H_6_). ^13^C NMR (100 MHz, DMSO-d_6_) δ_ppm_: 56.4 (2 × OMe), 60.4 (OMe), 67.0 (C_7_), 106.2 (C_10_, C_14_), 128.4 (C_9_), 138.3 (C_3_, C_5_), 143.3 (C_12_), 150.9 (C_2_, C_6_), 153.0 (C_11_, C_13_), 188.2 (C_8_). Anal. Calcd. for C_15_H_17_BrN_2_O_2_: C, 48.80; H, 4.64; N, 7.59. Found: C, 48.90; H, 4.58; N, 7.68.

##### 1-(2-(3,5-Dimethoxyphenyl)-2-oxoethyl)pyrazin-1-ium Bromide **3b**

Beige solid; 48% yield; mp 200–202 °C; IR ν(cm^−1^): 3086, 3009, 2978, 1694, 1601, 1454, 1420, 1346, 1304, 1194, 1180, 1161, 1018; ^1^H NMR (500 MHz, DMSO-d_6_) δ_ppm:_ 3.86 (s, 6H, 2 × OMe), 6.64 (s, 2H, H_7_), 6.94 (t, *J* = 2.0 Hz, 1H, H_12_), 7.19 (d, *J* = 2.5 Hz, 2H, H_10_, H_14_), 9.20 (d, *J* = 4.0 Hz, 2H, H_3_, H_5_), 9.71 (d, *J* = 4.0 Hz, 2H, H_2_, H_6_). ^13^C NMR (125 MHz, DMSO-d_6_) δ_ppm_: 55.8 (2 × OMe), 67.1 (C_7_), 106.2 (C_10_, C_14_), 106.2 (C_12_), 135.1 (C_9_), 138.4 (C_3_, C_5_), 150.9 (C_2_, C_6_), 160.9 (C_11_, C_13_), 189.1 (C_8_). Anal. Calcd. for C_14_H_15_BrN_2_O_3_: C, 49.57; H, 4.46; N, 8.26. Found: C, 49.55; H, 4.58; N, 8.28.

##### 1-(2-(3,4-Dimethoxyphenyl)-2-oxoethyl)pyrazin-1-ium Bromide **3c**

Beige solid; 52% yield; mp 196–198 °C; IR ν(cm^−1^): 3092, 3009, 2918, 1678, 1591, 1518, 1449, 1350, 1279, 1213, 1177, 1134; ^1^H NMR (500 MHz, DMSO-d_6_) δ_ppm_: 3.85 (s, 3H, OMe), 3.91 (s, 3H, OMe), 6.58 (s, 2H, H_7_), 7.23 (d, *J* = 8.5 Hz, 1H, H_13_), 7.52 (d, *J* = 1.5 Hz, 1H, H_10_), 7.78 (dd, *J* = 8.5; 1.5 Hz, 1H, H_14_), 9.18 (d, *J* = 3.0 Hz, 2H, H_3_, H_5_). 9.69 (d, *J* = 3.0 Hz, 2H, H_2_, H_6_). ^13^C NMR (125 MHz, DMSO-d_6_) δ_ppm_: 55.8 (OMe), 56.1 (OMe), 66.7 (C_7_), 110.4 (C_10_), 111.3 (C_13_), 123.6 (C_14_), 125.9 (C_9_), 138.4 (C_3_, C_5_), 148.9 (C_11_), 150.8 (C_2_, C_6_), 154.5 (C_12_), 187.5 (C_8_). Anal. Calcd. for C_14_H_15_BrN_2_O_3_: C, 49.57; H, 4.46; N, 8.26. Found: C, 49.63; H, 4.39; N, 8.30.

##### 1-(2-(4-Bromophenyl)-2-oxoethyl)pyrazin-1-ium Bromide **3d**

White solid; 57% yield; mp 240–243 °C; IR ν(cm^−1^): 3023, 2915, 1690, 1586, 1447, 1397, 1240, 986; ^1^H NMR (500 MHz, DMSO-d_6_) δ_ppm_: 6.57 (s, 2H, H_7_), 7.91 (d, *J* = 8.4 Hz, 2H, H_11_, H_13_), 8.02 (d, *J* = 8.4 Hz, 2H, H_10_, H_14_), 9.15 (d, *J* = 4.5 Hz, 2H, H_3_, H_5_), 9.70 (d, *J* = 4.4 Hz, 2H, H_2_, H_6_). ^13^C NMR (100 MHz, DMSO-d_6_) δ_ppm_: 66.8 (C_7_), 129.0 (C_12_), 130.2 (C_10_, C_14_), 132.2 (C_11_, C_13_, C_9_), 138.3 (C_3_, C_5_), 150.7 (C_2_, C_6_), 188.6 (C_8_). Anal. Calcd. for C_12_H_10_Br_2_N_2_O: C, 40.26; H, 2.82; N, 7.82. Found: C, 40.22; H, 2.78; N, 7.88.

##### Ethyl 6-(3,4,5-Trimethoxybenzoyl)pyrrolo[1,2-a]pyrazine-8-carboxylate **4a**

Beige solid; 40% yield; mp 162–164 °C; IR ν(cm^−1^): 2999, 2947, 1713, 1632, 1581, 1518, 1469, 1323, 1221, 1204; ^1^H NMR (400 Mz, CDCl_3_) δ_ppm_: 1.44 (t, *J* = 7.2 Hz, 3H, CH_3_), 3.91 (s, 6H, 2 × OMe), 3.97 (s, 3H, OMe), 4.45 (q, *J* = 7.2 Hz, 2H, CH_2_), 7.10 (s, 2H, H_12_, H_16_),7.91 (s, 1H, H_7_), 8.15 (d, *J* = 4.8 Hz, 1H, H_4_), 9.59 (dd, *J* = 4.8; 1.6 Hz, 1H, H_3_), 9.77 (d, *J* = 1.6 Hz, 1H, H_1_). ^13^C NMR (100 MHz, CDCl_3_) δ_ppm_: 14.6 (CH_3_), 56.6 (2 × OMe), 61.0 (CH_2_), 61.2 (OMe), 106.9 (C_12_, C_16_), 120.6 (C_3_), 123.3 (C_8_), 127.3 (C_7_), 132.1 (C_6_), 132.8 (C_4_), 134.0 (C_11_), 136.2 (C_9_), 142.1 (C_14_), 146.1 (C_1_), 153.3 (C_13_, C_15_), 163.2 (COO), 185.5 (C_10_). Anal. Calcd. for C_20_H_20_N_2_O_6_: C, 62.49; H, 5.24; N, 7.29. Found: C, 62.44; H, 5.20; N, 7.30.

##### Ethyl 6-(3,5-Dimethoxybenzoyl)pyrrolo[1,2-a]pyrazine-8-carboxylate **4b**

Beige solid; 35% yield; mp 160–162 °C; IR ν(cm^−1^): 2984, 2940, 1691, 1630, 1586, 1516, 1468, 1340, 1315, 1219; ^1^H NMR (400 Mz, CDCl_3_) δ_ppm_: 1.43 (t, *J* = 7.0 Hz, 3H, CH_3_), 3.86 (s, 6H, 2 × OMe), 4.40 (q, *J* = 7.0 Hz, 2H, CH_2_), 6.69 (t, *J* = 2.5 Hz, 1H, H_14_), 6.94 (d, *J* = 2.5 Hz, 2H, H_12_, H_16_), 7.91 (s, 1H, H_7_), 8.15 (d, *J* = 5.0 Hz, 1H, H_4_), 9.63 (dd, *J* = 4.5; 1.5 Hz, 1H, H_3_), 9.77 (bs, 1H, H_1_). ^13^C NMR (100 MHz, CDCl_3_) δ_ppm_: 14.6 (CH_3_), 56.8 (2 × OMe), 61.0 (CH_2_), 104.5 (C_14_), 107.1 (C_12_, C_16_), 109.3 (C_6_), 120.7 (C_3_), 123.3 (C_8_), 127.7 (C_7_), 132.2 (C_9_), 132.9 (C_4_), 140.1 (C_11_), 146.1 (C_1_), 160.9 (C_13_, C_15_), 163.2 (COO), 186.1 (C_10_). Anal. Calcd. for C_19_H_18_N_2_O_5_: C, 64.40; H, 5.12; N, 7.91. Found: C, 64.44; H, 5.15; N, 7.89.

##### Ethyl 6-(3,4-Dimethoxybenzoyl)pyrrolo[1,2-a]pyrazine-8-carboxylate **4c**

Beige solid; 30% yield; mp 170–172°C; IR ν(cm^−1^): 2926, 2853, 1705, 1674, 1663, 1595, 1466, 1269, 1209, 1136, 1022; ^1^H NMR (500 Mz, CDCl_3_) δ_ppm_: 1.43 (t, *J* = 7.0 Hz, 3H, CH_3_), 3.96 (s, 3H, OMe), 3.98 (s, 3H, OMe), 4.42 (q, *J* = 7.0 Hz, 2H, CH_2_), 6.97 (d, *J* = 8.5 Hz, 1H, H_15_), 7.44 (s, 1H, H_12_), 7.50 (d, *J* = 8.0 Hz, 1H, H_16_), 7.88 (s, 1H, H_7_), 8.10 (d, *J* = 4.0 Hz, 1H, H_4_), 9.54 (d, *J* = 4.0 Hz, 1H, H_3_), 9.74 (s, 1H, H_1_). ^13^C NMR (125 MHz, CDCl_3_) δ_ppm_: 14.5 (CH_3_), 56.1 (OMe), 56.2 (OMe), 60.8 (CH_2_), 110.2 (C_15_), 111.6 (C_12_), 120.4 (C_3_), 123.4 (C_8_), 123.9 (C_16_), 126.9 (C_7_), 131.4 (C_11_), 131.8 (C_6_), 132.4 (C_4_), 134.5 (C_9_), 145.9 (C_1_), 149.3 (C_13_), 153.0 (C_14_), 163.2 (COO), 184.9 (C_10_). Anal. Calcd. for C_19_H_18_N_2_O_5_: C, 64.40; H, 5.12; N, 7.91. Found: C, 64.45; H, 5.09; N, 7.92.

##### Ethyl 6-(4-Bromobenzoyl)pyrrolo[1,2-a]pyrazine-8-carboxylate **4d**

Beige solid; 35% yield; mp 167–169 °C; IR ν(cm^−1^): 2924, 2866, 1699, 1630, 1587, 1522, 1466, 1350, 1269, 1225; ^1^H NMR (400 Mz, CDCl_3_) δ_ppm_: 1.44 (t, *J* = 7.2 Hz, 3H, CH_3_), 4.44 (q, *J* = 7.2 Hz, 2H, CH_2_), 7.69 (d, *J* = 8.4 Hz, 2H, H_12_, H_16_), 7.73 (d, *J* = 8.4 Hz, 2H, H_13_, H_15_), 7.83 (s, 1H, H_7_), 8.17 (d, *J* = 4.4 Hz, 1H, H_4_), 9.64 (dd, *J* = 4.8; 1.6 Hz, 1H, H_3_), 9.78 (d, 1H, *J* = 1.2 Hz, H_1_). ^13^C NMR (100 MHz, CDCl_3_) δ_ppm_: 14.5 (CH_3_), 60.9 (CH_2_), 120.5 (C_3_), 122.9 (C_8_), 127.3 (C_14_), 127.4 (C_7_), 130.6 (C_13_, C_15_), 132.0 (C_12_, C_16_), 132.1 (C_6_), 132.9 (C_4_), 134.5 (C_9_), 137.5 (C_11_, C_9_), 145.9 (C_1_), 163.0 (COO), 185.1 (C_10_). Anal. Calcd. for C_17_H_13_BrN_2_O_3_: C, 54.71; H, 3.51; N, 7.51. Found: C, 54.75; H, 4.48; N, 7.49.

##### 4-(4-Chlorophenyl)-1-(2-oxo-2-(3,4,5-trimethoxyphenyl)ethyl)pyrimidin-1-ium Bromide **6a**

Yellow solid; 53% yield; mp 205–207 °C; IR ν(cm^−1^): 3009, 2997, 2958, 1674, 1630, 1593, 1452, 1416, 1339, 1314, 1198, 1165, 1128, 1093; ^1^H NMR (400 MHz, DMSO-d_6_) δ_ppm_: 3.82 (s, 3H, OMe), 3.92 (s, 6H, 2 × OMe), 6.53 (s, 2H, H_7_), 7.42 (s, 2H, H_10_, H_14_), 7.81 (d, *J* = 8.4 Hz, 2H, H_17_, H_19_), 8.51 (d, *J* = 8.4 Hz, 2H, H_16_, H_20_), 9.04 (d, *J* = 6.8 Hz, 1H, H_5_), 9.43 (dd, *J* = 6.8; 0.8 Hz, 1H, H_6_), 9.81 (s, 1H, H_1_). ^13^C NMR (100 MHz, DMSO-d_6_) δ_ppm_: 56.3 (2 × OMe), 60.3 (OMe), 62.4 (CH_2_), 106.0 (C_10_, C_14_), 118.2 (C_5_), 128.5 (C_9_), 129.9 (C_17_, C_19_), 130.9 (C_16_, C_20_), 131.7 (C_15_), 139.9 (C_18_), 143.1 (C_12_), 153.0 (C_11_, C_13_), 153.2 (C_6_), 154.5 (C_2_), 167.6 (C_4_), 189.1 (C_8_). Anal. Calcd. for C_21_H_20_BrClN_2_O_4_: C, 52.57; H, 4.20; N, 5.84. Found: C, 51.55; H, 4.18; N, 5.88.

##### 4-(4-Chlorophenyl)-1-(2-(3,5-dimethoxyphenyl)-2-oxoethyl)pyrimidin-1-ium Bromide **6b**

Yellow solid; 53% yield; mp 200–203 °C; IR ν(cm^−1^): 3026, 2988, 2930, 1692, 1628, 1593, 1549, 1454, 1341, 1316, 1298, 1206, 1155, 1009; ^1^H NMR (500 MHz, DMSO-d_6_) δ_ppm_: 3.87 (s, 6H, 2 × OMe), 6.40 (s, 2H, H_7_), 6.95 (t, *J* = 2.0 Hz, 1H, H_12_), 7.21 (d, *J* = 2.5 Hz, 2H, H_10_, H_14_), 7.81 (d, *J* = 8.5 Hz, 2H, H_17_, H_19_), 8.50 (d, *J* = 8.5 Hz, 2H, H_16_, H_20_), 8.99 (d, *J* = 7.0 Hz, 1H, H_5_), 9.36 (dd, *J* = 7.0; 1.5 Hz, 1H, H_6_), 9.74 (s, 1H, H_1_). ^13^C NMR (125 MHz, DMSO-d_6_) δ_ppm_: 55.8 (2 × OMe), 62.6 (CH_2_), 106.2 (C_10_, C_14_), 106.3 (C_12_), 118.3 (C_5_), 130.0 (C_17_, C_19_), 131.1 (C_16_, C_20_), 131.8 (C_15_), 135.3 (C_9_), 140.0 (C_18_), 153.3 (C_6_), 154.5 (C_2_), 149.0 (C_11_, C_13_), 167.7 (C_4_), 190.0 (C_8_). Anal. Calcd. for C_20_H_18_BrClN_2_O_3_: C, 53.41; H, 4.03; N, 6.23. Found: C, 53.40; H, 4.00; N, 6.19.

##### 4-(4-Chlorophenyl)-1-(2-(3,4-dimethoxyphenyl)-2-oxoethyl)pyrimidin-1-ium Bromide **6c**

Yellow solid; 53% yield; mp 208–210 °C; IR ν(cm^−1^): 3028, 2916, 1670, 1634, 1593, 1348, 1275, 1177, 1136, 1019, 839; ^1^H NMR (500 MHz, DMSO-d_6_) δ_ppm_: 3.86 (s, 3H, OMe), 3.92 (s, 3H, OMe), 6.38 (s, 2H, H_7_), 7.24 (d, *J* = 8.5 Hz, 1H, H_13_), 7.54 (d, *J* = 2.0 Hz, 1H, H_10_), 7.80–7.82 (m, 3H, H_17,_ H_19,_ H_14_), 8.50 (d, *J* = 8.5 Hz, 2H, H_16_, H_20_), 8.98 (d, *J* = 6.5 Hz, 1H, H_5_), 9.36 (dd, *J* = 7.0; 1.0 Hz, 1H, H_6_), 9.75 (s, 1H, H_1_). ^13^C NMR (125 MHz, DMSO-d_6_) δ_ppm_: 55.8 (OMe), 56.1 (OMe), 62.2 (CH_2_), 110.3 (C_10_), 111.3 (C_13_), 118.2 (C_5_), 123.6 (C_14_), 126.1 (C_9_), 130.0 (C_17_, C_19_), 131.0 (C_16_, C_20_), 131.8 (C_15_), 140.0 (C_18_), 149.0 (C_11_), 153.4 (C_6_), 154.4 (C_12_), 154.5 (C_2_), 167.7 (C_4_), 188.5 (C_8_). Anal. Calcd. for C_20_H_18_BrClN_2_O_3_: C, 53.41; H, 4.03; N, 6.23. Found: C, 53.45; H, 4.02; N, 6.28.

##### 1-(2-(4-Bromophenyl)-2-oxoethyl)-4-(4-chlorophenyl)pyrimidin-1-ium Bromide **6d**

Yellow solid; 55% yield; mp 240–243 °C; IR ν(cm^−1^): 2990, 2918, 1688, 1626, 1584, 1449, 1208, 981; ^1^H NMR (500 MHz, DMSO-d_6_) δ_ppm_: 6.41 (s, 2H, H_7_), 7.82 (d, *J* = 8.5 Hz, 2H, H_17,_ H_19_), 7.92 (d, *J* = 8.5 Hz, 2H, H_11_, H_13_), 8.04 (d, *J* = 8.5 Hz, 2H, H_10_, H_14_), 8.50 (d, *J* = 8.5 Hz, 2H, H_16_, H_20_), 8.99 (d, *J* = 7.0 Hz, 1H, H_5_), 9.35 (d, *J* = 7.0 Hz, 1H, H_6_), 9.73 (s, 1H, H_1_). ^13^C NMR (125 MHz, DMSO-d_6_) δ_ppm_: 62.4 (CH_2_), 118.3 (C_5_), 129.0 (C_12_), 130.0 (C_17_, C_19_), 130.3 (C_10_, C_14_), 131.0 (C_16_, C_20_), 131.8 (C_15_), 132.4 (C_11_, C_13_), 135.5 (C_9_), 140.0 (C_18_), 153.4 (C_6_), 154.5 (C_2_), 167.7 (C_4_), 189.6 (C_8_). Anal. Calcd. for C_18_H_13_Br_2_ClN_2_O: C, 46.14; H, 2.80; N, 5.98. Found: C, 46.15; H, 2.80; N, 6.00.

##### Ethyl 3-(4-Chlorophenyl)-7-(3,4,5-trimethoxybenzoyl)pyrrolo[1,2-c]pyrimidine-5-carboxylate **7a**

Brown solid; 36% yield; mp 180–181 °C; IR ν(cm^−1^): 3088, 2936, 1705, 1622, 1584, 1478, 1412, 1340, 1209, 1128; ^1^H NMR (400 MHz, CDCl_3_) δ_ppm_: 1.43 (t, *J* = 7.2 Hz, 3H, CH_3_), 3.94 (s, 6H, 2 × OMe), 3.97 (s, 3H, OMe), 4.42 (q, *J* = 7.2 Hz, 2H, CH_2_), 7.12 (s, 2H, H_18_, H_22_), 7.50 (d, *J* = 8.0 Hz, 2H, H_12_, H_14_),7.89 (s, 1H, H_6_), 8.15 (d, *J* = 8.5 Hz, 2H, H_11_, H_15_), 8.61 (s, 1H, H_4_), 10.53 (s, 1H, H_1_). ^13^C NMR (100 MHz, CDCl_3_) δ_ppm_: 14.7 (CH_3_), 56.6 (2 × OMe), 60.7 (CH_2_), 61.2 (OMe), 106.8 (C_18_, C_22_), 108.7 (C_4_), 122.4 (C_5_), 128.3 (C_11_, C_15_), 129.4 (C_12_, C_14_), 129.5 (C_17_), 129.7 (C_6_), 134.1 (C_7_), 135.2 (C_10_), 136.4 (C_13_), 140.7 (C_8_), 141.0 (C_1_), 142.0 (C_20_), 148.6 (C_3_), 153.3 (C_19_, C_21_), 163.7 (COO), 184.5 (C_16_). Anal. Calcd. for C_26_H_23_ClN_2_O_6_: C, 63.10; H, 4.68; N, 5.66. Found: C, 63.14; H, 4.70; N, 5.69.

##### Ethyl 3-(4-Chlorophenyl)-7-(3,5-trimethoxybenzoyl)pyrrolo[1,2-c]pyrimidine-5-carboxylate **7b**

Yellow solid; 40% yield; mp 222–225 °C; IR ν(cm^−1^): 3088, 2982, 2936, 1690, 1620, 1593, 1524, 1474, 1350, 1207, 1161, 1049; ^1^H NMR (500 Mz, CDCl_3_) δ_ppm_: 1.43 (t, *J* = 7.0 Hz, 3H, CH_3_), 3.87 (s, 6H, 2 × OMe), 4.42 (q, *J* = 7.0 Hz, 2H, CH_2_), 6.70 (t, *J* = 2.0 Hz, 1H, H_20_), 6.97 (d, *J* = 2.0 Hz, 2H, H_18_, H_22_), 7.49 (d, *J* = 8.5 Hz, 2H, H_12_, H_14_), 7.89 (s, 1H, H_6_), 8.12 (d, *J* = 8.5 Hz, 2H, H_11_, H_15_), 8.62 (d, *J* = 1.5 Hz, 1H, H_4_), 10.58 (d, *J* = 1.0 Hz, 1H, H_1_). ^13^C NMR (125 MHz, CDCl_3_)δ_ppm_: 14.7 (CH_3_), 55.8 (2 × OMe), 60.7 (CH_2_),104.3 (C_20_), 107.1 (C_18_, C_22_), 107.4 (C_5_), 108.7 (C_4_), 122.5 (C_7_), 128.3 (C_11_, C_15_), 129.4 (C_12_, C_14_), 130.1 (C_6_), 135.2 (C_10_, C_17_), 136.4 (C_13_), 140.9 (C_8_), 141.0 (C_1_), 148.8 (C_3_), 160.9 (C_19_, C_21_), 163.7 (COO), 185.2 (C_16_). Anal. Calcd. for C_26_H_23_ClN_2_O_6_: C, 63.10; H, 4.68; N, 5.66. Found: C, 63.12; H, 4.65; N, 5.64.

##### Ethyl 3-(4-Chlorophenyl)-7-(3,4-trimethoxybenzoyl)pyrrolo[1,2-*c*]pyrimidine-5-carboxylate **7c**

Beige solid; 35% yield; mp 150–153 °C; IR ν(cm^−1^): 2976, 2930, 1703, 1620, 1516, 1476, 1267, 1211, 1175, 1089; ^1^H NMR (500 Mz, CDCl_3_) δ_ppm_: 1.43 (t, *J* = 7.0 Hz, 3H, CH_3_), 3.98 (s, 3H, OMe), 4.00 (s, 3H, OMe), 4.42 (q, *J* = 7.0 Hz, 2H, CH_2_), 6.99 (d, *J* = 8.0 Hz, 1H, H_21_), 7.44–7.54 (m, 6H, H_18_, H_22_, H_12_, H_14_, H_22_, H_18_), 7.87 (s, 1H, H_6_), 8.12 (d, *J* = 8.0 Hz, 2H, H_11_, H_15_), 8.60 (s, 1H, H_4_), 10.52 (s, 1H, H_1_). ^13^C NMR (125 MHz, CDCl_3_) δ_ppm_: 14.7 (CH_3_), 56.2 (OMe), 56.3 (OMe), 60.7 (CH_2_), 107.1 (C_5_), 108.7 (C_4_), 110.3 (C_21_), 111.7 (C_18_), 122.7 (C_7_), 123.8 (C_22_), 128.3 (C_11_, C_15_), 129.4 (C_12_, C_14_), 129.5 (C_6_), 131.6 (C_17_), 135.3 (C_10_), 136.3 (C_13_), 140.5 (C_8_), 141.0 (C_1_), 148.4 (C_3_), 149.4 (C_19_), 153.0 (C_20_), 163.8 (COO), 184.3 (C_16_). Anal. Calcd. for C_26_H_23_ClN_2_O_6_: C, 63.10; H, 4.68; N, 5.66. Found: C, 63.13; H, 4.69; N, 5.60.

##### Ethyl 7-(4-Bromobenzoyl)-3-(4-chlorophenyl)pyrrolo[1,2-*c*]pyrimidine-5-carboxylate **7d**

Yellow solid; 40% yield; mp 162–164 °C; IR ν(cm^−1^): 3094, 2986, 1722, 1699, 1613, 1471, 1265, 1234, 1202; ^1^H NMR (500 Mz, CDCl_3_) δ_ppm_: 1.43 (t, *J* = 7.0 Hz, 3H, CH_3_), 4.42 (q, *J* = 7.0 Hz, 2H, CH_2_), 7.49 (d, *J* = 8.0 Hz, 2H, H_12_, H_14_), 7.70 (d, *J* = 8.0 Hz, 2H, H_18_, H_22_), 7.74 (d, *J* = 8.5 Hz, 2H, H_19_, H_21_), 7.81 (s, 1H, H_6_), 8.13 (d, *J* = 8.5 Hz, 2H, H_11_, H_15_), 8.63 (s, 1H, H_4_), 10.58 (s, 1H, H_1_). ^13^C NMR (125 MHz, CDCl_3_) δ_ppm_: 14.7 (CH_3_), 60.8 (CH_2_), 107.5 (C_5_), 108.7 (C_4_), 122.2 (C_7_), 127.2 (C_20_), 128.3 (C_11_, C_15_), 129.4 (C_12_, C_14_), 129.9 (C_6_), 130.7 (C_19_, C_21_), 132.1 (C_18_, C_22_), 135.1 (C_10_), 137.8 (C_17_), 136.5 (C_13_), 140.2 (C_8_), 141.0 (C_1_), 149.0 (C_3_), 163.6 (COO), 184.2 (C_16_). Anal. Calcd. for C_23_H_16_BrClN_2_O_3_: C, 57.11; H, 3.33; N, 5.79. Found: C, 57.13; H, 3.39; N, 5.76.

##### 1-(2-oxo-2-(3,4,5-Trimethoxyphenyl)ethyl)quinolin-1-ium Bromide **9a**

Yellow solid; 80% yield; mp 278–279 °C; IR ν(cm^−1^): 3064, 2964, 1693, 1589, 1416, 1320, 1118, 995; ^1^H NMR (500 MHz, DMSO-d_6_) δ_ppm_: 3.96 (s, 3H, OMe), 4.00 (s, 6H, 2 × OMe), 7.58 (s, 2H, H_14_, H_18_), 7.67 (s, 2H, H_11_), 7.96 (t, *J* = 7.0 Hz, 1H, H_6_), 8.06–8.15 (overlapped signals, 3H, H_7_, H_3_, H_5_), 8.27 (d, *J* = 8.5 Hz, 1H, H_8_), 9.00 (d, *J* = 7.5 Hz, 1H, H_4_), 10.34 (d, *J* = 6.0 Hz, 1H, H_2_). ^13^C NMR (125 MHz, DMSO-d_6_) δ_ppm_: 57.1 (2 × OMe), 61.2 (OMe), 64.3 (C_11_), 106.7 (C_14_, C_18_), 119.0 (C_5_), 122.1 (C_3_), 128.6 (C_13_), 129.9 (C_9_), 130.4 (C_6_), 130.6 (C_8_), 136.3 (C_7_), 139.5 (C_10_), 144.4 (C_1_), 147.5 (C_4_), 151.3 (C_2_), 153.7 (C_15_, C_17_), 189.1 (C_12_). Anal. Calcd. for C_20_H_20_BrNO_4_: C, 57.43; H, 4.82; N, 3.35. Found: C, 57.42; H, 4.80; N, 3.36.

##### 1-(2-oxo-2-(3,5-Trimethoxyphenyl)ethyl)quinolin-1-ium Bromide **9b**

Beige solid; 63% yield; mp 265–268 °C; IR ν(cm^−1^): 3007, 2871, 1680, 1649, 1597, 1483, 1352, 1290, 1203, 1166, 1022; ^1^H NMR (500 MHz, DMSO-d_6_) δ_ppm_: 3.87 (s, 6H, 2 × OMe), 6.97 (s, 1H, H_16_), 7.01 (s, 2H, H_11_), 7.28 (s, 2H, H_14_, H_18_), 8.07 (t, *J* = 7.0 Hz, 1H, H_6_), 8.24 (t, *J* = 7.5 Hz, 1H, H_7_), 8.33 (t, *J* = 6.5 Hz, 1H, H_3_), 8.40 (d, *J* = 8.5 Hz, 1H, H_5_), 8.55 (d, *J* = 7.5 Hz, 1H, H_8_), 9.45 (d, *J* = 7.5 Hz, 1H, H_4_), 9.50 (d, *J* = 5.0 Hz, 1H, H_2_). ^13^C NMR (125 MHz, DMSO-d_6_) δ_ppm_: 55.8 (2 × OMe), 63.3 (C_11_), 106.5 (C_14_, C_18_, C_16_) 119.1 (C_5_), 122.2 (C_3_), 129.4 (C_9_), 130.0 (C_6_), 130.7 (C_8_), 135.4 (C_13_), 136.0 (C_7_), 138.6 (C_10_), 148.7 (C_4_), 151.0 (C_2_), 160.8 (C_15_, C_17_), 190.6 (C_12_). Anal. Calcd. for C_19_H_8_BrNO_3_: C, 58.78; H, 4.67; N, 3.61. Found: C, 58.77; H, 4.65; N, 3.63.

##### 1-(2-oxo-2-(3,4-Trimethoxyphenyl)ethyl)quinolin-1-ium bromide **9c**

Beige solid; 51% yield; mp 270–271 °C; IR ν(cm^−1^): 3042, 3010, 2934, 1676, 1644, 1592, 1515, 1280, 1160, 1021; ^1^H NMR (500 MHz, DMSO-d_6_) δ_ppm_: 3.86 (s, 3H, OMe), 3.93 (s, 3H, OMe), 7.00 (s, 2H, H_11_), 7.27 (d, *J* = 8.5 Hz, 1H, H_17_), 7.58 (d, *J* = 1.5 Hz, 1H, H_14_), 7.89 (dd, *J* = 8.5; 1.5 Hz, 1H, H_18_), 8.07 (t, *J* = 7.5 Hz, 1H, H_6_), 8.23 (t, *J* = 7.5 Hz, 1H, H_7_), 8.32 (dd, *J* = 8.5; 6.0 Hz, 1H, H_3_), 8.38 (d, *J* = 8.0 Hz, 1H, H_5_), 8.55 (d, *J* = 8.0 Hz, 1H, H_8_), 9.45 (d, *J* = 8.5 Hz, 1H, H_4_), 9.53 (d, *J* = 5.5 Hz, 1H, H_2_). ^13^C NMR (125 MHz, DMSO-d_6_) δ_ppm_: 55.8 (OMe), 56.1 (OMe), 62.9 (C_11_), 110.6 (C_14_), 111.3 (C_17_), 119.1 (C_5_), 122.2 (C_3_), 123.8 (C_18_), 126.3 (C_13_), 129.4 (C_9_), 130.0 (C_6_), 130.7 (C_8_), 136.0 (C_7_), 138.6 (C_10_), 148.5 (C_4_), 148.9 (C_15_), 151.0 (C_2_), 154.5 (C_16_), 189.0 (C_12_). Anal. Calcd. for C_19_H_8_BrNO_3_: C, 58.78; H, 4.67; N, 3.61. Found: C, 58.76; H, 4.67; N, 3.62.

##### 1-(2-(4-Bromophenyl)-2-oxoethyl)quinolin-1-ium Bromide **9d**

Physical and spectral data are similar to the ones reported in ref. [6].

##### Ethyl 1-(3,4,5-Trimethoxybenzoyl)pyrrolo[1,2-*a*]quinoline-3-carboxylate 1**0a**

Beige solid; 39% yield; mp 231–234 °C; IR ν(cm^−1^): 2947, 1711, 1628, 1584, 1459, 1361, 1192, 1144, 759; ^1^H NMR (500 Mz, CDCl_3_) δ_ppm_: 1.41 (t, *J* = 7.0 Hz, 3H, CH_2_CH_3_), 3.93 (s, 6H, 2 × OMe), 3.99 (s, 3H, OMe), 4.39 (q, *J* = 7.0 Hz, 2H, CH_2_CH_3_), 7.36 (s, 2H, H_15_,H_19_), 7.49 (t, *J* = 7.5 Hz, 1H, H_7_), 7.57 (dt, *J* = 7.5; 1.5 Hz, 1H, H_8_), 7.66 (s, 1H, H_2_), 7.69 (d, *J* = 9.0 Hz, 1H, H_5_), 7.82 (d, *J* = 7.5 Hz, 1H, H_6_), 8.00 (d, *J* = 8.5 Hz, 1H, H_9_), 8.34 (d, *J* = 9.0 Hz, 1H, H_4_). ^13^C NMR (125 MHz, CDCl_3_) δ_ppm_: 14.7 (CH_2_CH_3_), 56.5 (2 × OMe), 60.4 (CH_2_CH_3_), 61.2 (OMe), 107.8 (C_15_, C_19_), 107.9 (C_3_), 117.9 (C_4_), 120.2 (C_9_), 125.3 (C_11_), 125.6 (C_7_), 128.0 (C_1_), 128.8 (C_2_), 128.9 (C_5_), 129.0 (C_8_), 129.1 (C_6_), 133.3 (C_12_), 133.6 (C_14_), 140.3 (C_10_), 142.6 (C_17_), 153.2 (C_16_, C_18_), 164.3 (COO), 184.5 (C_13_). Anal. Calcd. for C_25_H_23_NO_6_: C, 69.27; H, 5.35; N, 3.23. Found: C, 69.28; H, 5.33; N, 3.25. HRMS (HESI^+^): m/z calcd for C_25_H_23_NO_6_ (M^+^): 433.1525; found 433.1519.

##### Ethyl 1-(3,5-Trimethoxybenzoyl)pyrrolo[1,2-*a*]quinoline-3-carboxylate **10b**

Yellow solid; 43% yield; mp 240–241 °C; IR ν(cm^−1^): 2976, 1710, 1629, 1595, 1457, 1360, 1192, 760; ^1^H NMR (500 Mz, CDCl_3_) δ_ppm_: 1.40 (t, *J* = 7.0 Hz, 3H, CH_2_CH_3_), 3.88 (s, 6H, 2 × OMe), 4.38 (q, *J* = 7.0 Hz, 2H, CH_2_CH_3_), 6.75 (t, *J* = 2.0 Hz, 1H, H_17_), 7.22 (d, *J* = 2.5 Hz, 2H, H_15_, H_19_), 7.49 (t, *J* = 7.5 Hz, 1H, H_7_), 7.57 (t, *J* = 7.5 Hz, 1H, H_8_), 7.68 (s, 1H, H_2_), 7.69 (d, *J* = 9.0 Hz, 1H, H_5_), 7.82 (d, *J* = 7.0 Hz, 1H, H_6_), 8.04 (d, *J* = 9.0 Hz, 1H, H_9_), 8.33 (d, *J* = 9.0 Hz, 1H, H_4_). ^13^C NMR (125 MHz, CDCl_3_) δ_ppm_: 14.7 (CH_2_CH_3_), 55.8 (2 × OMe), 60.4 (CH_2_CH_3_), 105.6 (C_17_), 107.9 (C_3_), 108.9 (C_15_, C_19_), 117.9 (C_4_), 120.3 (C_9_), 125.3 (C_11_), 125.6 (C_7_), 128.2 (C_1_), 128.9 (C_5_), 129.0 (C_8_), 129.1 (C_6_), 129.7 (C_2_), 133.3 (C_12_), 140.5 (C_10_, C_14_), 160.9 (C_16_, C_18_), 164.3 (COO), 184.8 (C_13_). Anal. Calcd. for C_24_H_21_NO_5_: C, 71.45; H, 5.25; N, 3.47. Found: C, 71.43; H, 5.24; N, 3.49.

##### Ethyl 1-(3,4-Trimethoxybenzoyl)pyrrolo[1,2-*a*]quinoline-3-carboxylate **10c**

Beige solid; 42% yield; mp 259–261 °C; IR ν(cm^−1^): 2919; 1709, 1638, 1514, 1461, 1384, 1268, 1140, 1081, 1020; ^1^H NMR (500 Mz, CDCl_3_) δ_ppm_: 1.40 (t, *J* = 7.0 Hz, 3H, CH_2_CH_3_), 3.99 (s, 3H, OMe), 4.01 (s, 3H, OMe), 4.38 (q, *J* = 7.0 Hz, 2H, CH_2_CH_3_), 6.99 (d, *J* = 8.5 Hz, 1H, H_18_), 7.47 (dt, *J* = 8.0; 1.0 Hz, 1H, H_7_), 7.55 (dt, *J* = 7.0; 1.5 Hz, 1H, H_8_), 7.62 (s, 1H, H_2_), 7.65–7.69 (overlapped signals, 2H, H_5_, H_15_), 7.77 (dd, *J* = 8.0; 2.0 Hz, 1H, H_19_), 7.80 (dd, *J* = 8.0; 1.5 Hz, 1H, H_6_), 7.98 (d, *J* = 8.5 Hz, 1H, H_9_), 8.32 (d, *J* = 9.5 Hz, 1H, H_4_). ^13^C NMR (125 MHz, CDCl_3_) δ_ppm_: 14.7 (CH_2_CH_3_), 56.2 (OMe), 56.3 (OMe), 60.3 (CH_2_CH_3_), 107.6 (C_3_), 110.1 (C_18_), 112.0 (C_15_), 117.9 (C_4_), 120.1 (C_9_), 125.2 (C_11_), 125.5 (C_7_), 125.6 (C_19_), 128.1 (C_1_), 128.4 (C_2_), 128.6 (C_5_), 128.8 (C_8_), 129.1 (C_6_), 131.2 (C_14_), 133.3 (C_12_), 139.9 (C_10_), 149.3 (C_16_), 153.6 (C_17_), 164.4 (C_20_), 184.6 (C_13_). Anal. Calcd. for C_24_H_21_NO_5_: C, 71.45; H, 5.25; N, 3.47. Found: C, 71.46; H, 5.26; N, 3.48.

##### Ethyl 1-(4-Bromobenzoyl)pyrrolo[1,2-*a*]quinoline-3-carboxylate **10d**

Yellow solid; 60% yield; mp 179–180 °C; ^1^H NMR (CDCl_3_, 500 MHz) δ_ppm_: 1.40 (t, *J* = 7.0 Hz, 3H, CH_2_CH_3_), 4.38 (q, *J* = 7.0 Hz, 2H, CH_2_CH_3_), 7.50 (dt, *J* = 8.0; 1.0 Hz, 1H, H_7_), 7.59 (dt, *J* = 7.0; 1.5 Hz, 1H, H_8_), 7.62 (s, 1H, H_2_), 7.69–7.72 (overlapped signals, 3H, H_5_, H_16_,H_18_), 7.82 (dd, *J* = 8.0; 1.5 Hz, 1H, H_6_), 7.97 (d, *J* = 8.5 Hz, 2H, H_15_, H_19_), 8.03 (d, *J* = 8.5 Hz, 1H, H_9_), 8.34 (d, *J* = 9.0 Hz, 1H, H_4_). ^13^C NMR (CDCl_3_, 125 MHz) δ_ppm_: 14.7 (CH_2_CH_3_), 60.4 (CH_2_CH_3_), 108.1 (C_3_), 117.8 (C_4_), 120.3 (C_9_), 125.3 (C_11_), 125.7 (C_7_), 127.8 (C_1_), 128.1 (C_17_), 129.0 (C_8_), 129.1 (C_6_), 129.8 (C_2_), 129.4 (C_5_), 131.8 (C_15_, C_19_), 132.0 (C_16_, C_18_), 133.3 (C_14_), 137.4 (C_12_), 140.7 (C_10_), 164.1 (C_20_), 183.8 (C_13_). Anal. Calcd. for C_22_H_16_BrNO_3_: C, 62.57; H, 3.82; N, 3.32. Found: C, 62.56; H, 3.80; N, 3.33.

##### 4-(2-oxo-2-(3,4,5-Trimethoxyphenyl)ethyl)benzo[*f*]quinolin-4-ium Bromide **12a**

Beige solid; 54% yield; mp 212–214 °C; IR ν(cm^−1^): 3092, 2980, 2933, 1690, 1584, 1416, 1323, 1234, 1159, 1122, 1064, 993, 762; ^1^H NMR (500 MHz, DMSO-d_6_) δ_ppm_: 3.82 (s, 3H, OMe), 3.93 (s, 6H, 2 × OMe), 7.18 (s, 2H, H_11_), 7.50 (s, 2H, H_14_, H_18_), 7.99 (t, *J* = 7.0 Hz, 1H, H_7_), 8.05 (t, *J* = 7.0 Hz, 1H, H_6_), 8.27 (d, *J* = 9.5 Hz, 1H, H_10_), 8.31 (d, *J* = 8.0 Hz, 1H, H_8_), 8.48 (dd, *J* = 8.5; 6.0 Hz, 1H, H_3_), 8.66 (d, *J* = 10.0 Hz, 1H, H_9_), 9.18 (d, *J* = 8.5 Hz, 1H, H_5_), 9.50 (d, *J* = 5.5 Hz, 1H, H_2_), 10.28 (d, *J* = 8.5 Hz, 1H, H_4_). ^13^C NMR (125 MHz, DMSO-d_6_) δ_ppm_: 56.4 (2 × OMe), 60.4 (OMe), 64.0 (C_11_), 106.4 (C_14_, C_18_), 116.3 (C_10_), 122.9 (C_3_), 124.3 (C_5_), 127.9 (C_4a_), 128.1 (C_4b_), 128.7 (C_13_), 129.5 (C_8_), 130.2 (C_6_), 130.2 (C_7_), 130.9 (C_8a_), 138.2 (C_9_), 140.1 (C_10a_), 142.6 (C_4_), 143.3 (C_16_), 148.2(C_2_), 153.0 (C_15_, C_17_), 189.7 (C_12_). Anal. Calcd. for C_24_H_22_BrNO_4_: C, 61.55; H, 4.73; N, 2.99. Found: C, 61.54; H, 4.70; N, 3.00.

##### 4-(2-oxo-2-(3,5-Trimethoxyphenyl)ethyl)benzo[*f*]quinolin-4-ium Bromide **12b**

Yellow solid; 51% yield; mp 163–165 °C; IR ν(cm^−1^): 3062, 2978, 2903, 1692, 1593, 1423, 1344, 1304, 1200, 1155, 1061, 810, 750; ^1^H NMR (400 MHz, DMSO-d_6_) δ_ppm_: 3.89 (s, 6H, 2 × OMe), 6.98 (t, *J* = 2.0 Hz, 1H, H_16_), 7.10 (s, 2H, H_11_), 7.31 (d, *J* = 2.0 Hz, 2H, H_14_, H_18_), 8.01 (t, *J* = 7.2 Hz, 1H, H_7_), 8.06 (t, *J* = 7.0 Hz, 1H, H_6_), 8.29 (d, *J* = 9.6 Hz, 1H, H_10_), 8.32 (d, *J* = 8.4 Hz, 1H, H_8_), 8.48 (dd, *J* = 8.4; 6.0 Hz, 1H, H_3_), 8.66 (d, *J* = 9.6 Hz, 1H, H_9_), 9.18 (d, *J* = 8.4 Hz, 1H, H_5_), 9.49 (d, *J* = 6.0 Hz, 1H, H_2_), 10.28 (d, *J* = 8.8 Hz, 1H, H_4_). ^13^C NMR (100 MHz, DMSO-d_6_) δ_ppm_: 55.8 (2 × OMe), 63.9 (C_11_), 106.5 (C_14_, C_18_, C_16_), 116.2 (C_10_), 122.8 (C_3_), 124.2 (C_5_), 127.9 (C_4a_), 128.0 (C_4b_), 129.4 (C_8_), 130.1 (C_6_), 130.2 (C_7_), 130.8 (C_8a_), 135.4 (C_13_), 138.1 (C_9_), 140.0 (C_10a_), 142.5 (C_4_), 148.2 (C_2_), 160.8 (C_15_, C_17_), 190.5 (C_12_). Anal. Calcd. for C_23_H_20_BrNO_3_: C, 63.02; H, 4.60; N, 3.20. Found: C, 63.01; H, 4.59; N, 3.22.

##### 4-(2-oxo-2-(3,4-Trimethoxyphenyl)ethyl)benzo[*f*]quinolin-4-ium Bromide **12c**

Beige solid; 49% yield; mp 170–172 °C; IR ν(cm^−1^): 3069, 2974, 1684, 1591, 1508, 1420, 1344, 1260, 1150, 1013, 816, 764; ^1^H NMR (500 MHz, DMSO-d_6_) δ_ppm_: 3.87 (s, 3H, OMe), 3.94 (s, 3H, OMe), 7.09 (s, 2H, H_11_), 7.28 (d, *J* = 8.5 Hz, 1H, H_17_), 7.60 (s, 1H, H_14_), 7.91 (d, *J* = 8.0 Hz, 1H, H_18_), 7.99 (t, *J* = 7.0 Hz, 1H, H_7_), 8.05 (t, *J* = 7.0 Hz, 1H, H_6_), 8.26 (d, *J* = 9.5 Hz, 1H, H_10_), 8.30 (d, *J* = 7.5 Hz, 1H, H_8_), 8.47 (at, *J* = 7.0 Hz, 1H, H_3_), 8.65 (d, *J* = 9.5 Hz, 1H, H_9_), 9.18 (d, *J* = 8.0 Hz, 1H, H_5_), 9.50 (d, *J* = 5.0 Hz, 1H, H_2_), 10.27 (d, *J* = 8.5 Hz, 1H, H_4_). ^13^C NMR (125 MHz, DMSO-d_6_) δ_ppm_: 55.8 (OMe), 56.1 (OMe), 63.6 (C_11_), 110.6 (C_14_), 111.3 (C_17_), 116.3 (C_10_), 122.9 (C_3_), 123.9 (C_18_), 124.3 (C_5_), 126.3 (C_13_), 127.9 (C_4a_), 128.0 (C_4b_), 129.5 (C_8_), 130.1 (C_6_), 130.2 (C_7_), 130.9 (C_8a_), 138.1 (C_9_), 140.1 (C_10a_), 142.5 (C_4_), 148.3 (C_2_), 148.8 (C_15_), 154.5 (C_16_), 189.0 (C_12_). Anal. Calcd. for C_23_H_20_BrNO_3_: C, 63.02; H, 4.60; N, 3.20. Found: C, 63.03; H, 4.58; N, 3.21.

##### 4-(2-(4-Bromophenyl)-2-oxoethyl)benzo[*f*]quinolin-4-ium Bromide **12d**

Yellow solid; 50% yield; mp 246–248 °C; IR ν(cm^−1^): 3018, 2822, 1695, 1582, 1406, 1354, 1225, 991, 804, 754; ^1^H NMR (500 MHz, DMSO-d_6_) δ_ppm_: 7.05 (s, 2H, H_11_), 7.95 (d, *J* = 8.5 Hz, 2H, H_15_, H_17_), 8.00 (t, *J* = 7.0 Hz, 1H, H_7_), 8.06 (t, *J* = 7.0 Hz, 1H, H_6_), 8.09 (d, *J* = 8.5 Hz, 2H, H_14_, H_18_), 8.31 (d, *J* = 8.0 Hz, 1H, H_8_), 8.36 (d, *J* = 10.0 Hz, 1H, H_10_), 8.47 (dd, *J* = 8.5; 6.0 Hz, 1H, H_3_), 8.64 (d, *J* = 9.5 Hz, 1H, H_9_), 9.18 (d, *J* = 8.5 Hz, 1H, H_5_), 9.45 (d, *J* = 5.0 Hz, 1H, H_2_), 10.28 (d, *J* = 8.5 Hz, 1H, H_4_). ^13^C NMR (125 MHz, DMSO-d_6_) δ_ppm_: 63.8 (C_11_), 116.4 (C_10_), 122.9 (C_3_), 124.3 (C_5_), 127.9 (C_4a_), 128.0 (C_4b_), 129.0 (C_16_), 129.5 (C_8_), 130.1 (C_6_), 130.2 (C_7_), 130.6 (C_14_, C_18_), 130.9 (C_8a_), 132.2 (C_15_, C_17_), 132.7 (C_13_), 138.1 (C_9_), 140.2 (C_10a_), 142.6 (C_4_), 148.3 (C_2_), 190.2 (C_12_). Anal. Calcd. for C_21_H_15_BrNO: C, 66.86; H, 4.01; N, 3.71. Found: C, 66.85; H, 4.00; N, 3.73.

##### Ethyl 3-(3,4,5-Trimethoxybenzoyl)benzo[*f*]pyrrolo[1,2-*a*]quinoline-1-carboxylate **13a**

Yellow solid; 80% yield; mp 239–240 °C; IR ν(cm^−1^): 1703, 1632, 1579, 1499, 1416, 1344, 1227, 1128, 1078, 744; ^1^H NMR (500 Mz, CDCl_3_) δ_ppm_: 1.43 (t, *J* = 7.0 Hz, 3H, CH_2_CH_3_), 3.95 (s, 6H, 2 × OMe), 4.00 (s, 3H, OMe), 4.42 (q, *J* = 7.0 Hz, 2H, CH_2_CH_3_), 7.41 (s, 2H, H_15_, H_19_), 7.65 (t, *J* = 8.0 Hz, 1H, H_8_), 7.75 (dd, *J* = 8.5; 7.0 Hz, 1H, H_9_), 7.78 (s, 1H, H_2_), 7.96–7.98 (overlapped signals, 3H, H_7_, H_11_, H_12_), 8.55 (d, *J* = 9.5 Hz, 1H, H_6_), 8.62 (d, *J* = 9.5 Hz, 1H, H_5_), 8.64 (d, *J* = 8.5 Hz, 1H, H_10_). ^13^C NMR (125 MHz, CDCl_3_) δ_ppm_: 14.7 (CH_2_CH_3_), 56.6 (2 × OMe), 60.4 (CH_2_CH_3_), 61.2 (OMe), 107.2 (C_1_), 107.7 (C_15_, C_19_), 117.9 (C_6_), 119.7 (C_12_), 121.0 (C_10b_), 123.0 (C_5_), 123.8 (C_10_), 126.8 (C_8_), 127.6 (C_3_), 127.9 (C_9_), 129.0 (C_2_), 129.7 (C_7_), 129.8 (C_11_), 130.0 (C_10a_), 130.9 (C_6a_), 132.1 (C_14_), 133.6 (C_4a_), 140.3 (C_12a_), 142.6 (C_17_), 153.2 (C_16_, C_18_), 164.3 (COO), 183.9 (C_13_). Anal. Calcd. for C_29_H_25_NO_6_: C, 72.04; H, 5.21; N, 2.90. Found: C, 72.02; H, 5.20; N, 2.92.

##### Ethyl 3-(3,5-Dimethoxybenzoyl)benzo[*f*]pyrrolo[1,2-*a*]quinoline-1-carboxylate **13b**

Yellow solid; 70% yield; mp 246–247 °C; IR ν(cm^−1^): 1699, 1632, 1593, 1498, 1427, 1354, 1300, 1230, 1153, 1082, 744; ^1^H NMR (400 MHz, CDCl_3_) δ_ppm_: 1.42 (t, *J* = 7.2 Hz, 3H, CH_2_CH_3_), 3.88 (s, 6H, 2 × OMe), 4.40 (q, *J* = 7.2 Hz, 2H, CH_2_CH_3_), 6.76 (bs, 1H, H_17_), 7.26 (d, *J* = 2.0 Hz, 2H, H_15_, H_19_), 7.62 (t, *J* = 7.6 Hz, 1H, H_8_), 7.71 (t, *J* = 7.6 Hz, 1H, H_9_), 7.79 (s, 1H, H_2_), 7.93–7.98 (overlapped signals, 3H, H_7_, H_11_, H_12_), 8.51 (d, *J* = 9.6 Hz, 1H, H_6_), 8.56–8.60 (overlapped signals, 2H, H_5_, H_10_). ^13^C NMR (100 MHz, CDCl_3_) δ_ppm_: 14.6 (CH_2_CH_3_), 55.7 (2 × OMe), 60.2 (CH_2_CH_3_), 105.4 (C_17_), 107.1 (C_1_), 107.9 (C_15_, C_19_), 117.7 (C_6_), 119.7 (C_12_), 120.9 (C_10b_), 122.8 (C_5_), 123.7 (C_10_), 126.6 (C_8_), 127.6 (C_3_), 127.7 (C_9_), 128.8 (C_7_), 129.5 (C_11_), 129.8 (C_10a_), 130.2 (C_2_), 130.8 (C_6a_), 132.0 (C_4a_), 140.3 (C_12a_), 140.4 (C_14_), 160.8 (C_16_, C_18_), 164.1 (COO), 184.0 (C_13_). Anal. Calcd. for C_28_H_25_NO_5_: C, 74.16; H, 5.11; N, 3.09. Found: C, 74.15; H, 5.10; N, 3.11.

##### Ethyl 3-(3,4-Dimethoxybenzoyl)benzo[*f*]pyrrolo[1,2-*a*]quinoline-1-carboxylate **13c**

Yellow solid; 70% yield; mp 244–245 °C; IR ν(cm^−1^): 1699, 1630, 1512, 1427, 1344, 1232, 1137, 1078, 1022, 806, 740; ^1^H NMR (500 Mz, CDCl_3_) δ_ppm_: 1.43 (t, *J* = 7.0 Hz, 3H, CH_2_CH_3_), 4.00 (s, 3H, OMe), 4.03 (s, 3H, OMe), 4.41 (q, *J* = 7.0 Hz, 2H, CH_2_CH_3_), 7.02 (d, *J* = 8.0 Hz, 1H, H_19_), 7.63 (t, *J* = 7.5 Hz, 1H, H_8_), 7.69 (d, *J* = 2.0 Hz, 2H, H_15_), 7.73 (t, *J* = 7.0 Hz, 1H, H_9_), 7.75 (s, 1H, H_2_), 6.76 (dd, *J* = 8.0; 2.0 Hz, 1H, H_18_), 7.95–7.97 (overlapped signals, 3H, H_7_, H_11_, H_12_), 8.54 (d, *J* = 9.5 Hz, 1H, H_6_), 8.59 (d, *J* = 9.5 Hz, 1H, H_5_), 8.63 (d, *J* = 8.5 Hz, 1H, H_10_). ^13^C NMR (125 MHz, CDCl_3_) δ_ppm_: 14.7 (CH_2_CH_3_), 56.3 (OMe), 56.4 (OMe), 60.3 (CH_2_CH_3_), 107.0 (C_1_), 110.2 (C_19_), 112.0 (C_15_), 117.9 (C_6_), 119.7 (C_12_), 120.9 (C_10b_), 123.0 (C_10_), 123.5 (C_5_), 125.4 (C_18_), 126.7 (C_8_), 127.7 (C_3_), 127.8 (C_9_), 129.0 (C_7_), 129.2 (C_2_), 129.7 (C_11_), 130.0 (C_10a_), 130.9 (C_6a_), 131.2 (C_14_), 132.0 (C_4a_), 140.0 (C_12a_), 149.4 (C_16_), 153.5 (C_17_), 164.4 (COO), 184.0 (C_13_). Anal. Calcd. for C_28_H_25_NO_5_: C, 74.16; H, 5.11; N, 3.09. Found: C, 74.14; H, 5.10; N, 3.10.

##### Ethyl 3-(4-Bromobenzoyl)benzo[*f*]pyrrolo[1,2-*a*]quinoline-1-carboxylate **13d**

Yellow solid; 65% yield; mp 258–259 °C; IR ν(cm^−1^): 1708, 1636, 1502, 1425, 1339, 1231, 1078, 742; ^1^H NMR (500 Mz, CDCl_3_) δ_ppm_: 1.42 (t, *J* = 7.0 Hz, 3H, CH_2_CH_3_), 4.41 (q, *J* = 7.0 Hz, 2H, CH_2_CH_3_), 7.64 (t, *J* = 7.5 Hz, 1H, H_8_), 7.72–7.75 (overlapped signals, 4H, H_15_, H_19_, H_9_, H_2_), 7.96–7.97 (overlapped signals, 3H, H_7_, H_11_, H_12_), 8.00 (d, *J* = 8.5 Hz, 2H, H_15_, H_19_), 8.54 (d, *J* = 9.6 Hz, 1H, H_6_), 8.61–8.63 (overlapped signals, 2H, H_5_, H_10_). ^13^C NMR (125 MHz, CDCl_3_) δ_ppm_: 14.7 (CH_2_CH_3_), 60.4 (CH_2_CH_3_), 107.4 (C_1_), 117.8 (C_6_), 119.7 (C_12_), 121.1 (C_10b_), 123.0 (C_5_), 124.2 (C_10_), 126.9 (C_8_), 127.4 (C_17_), 127.9 (C_9_), 128.0 (C_3_), 129.0 (C_7_), 129.8 (C_11_), 129.9 (C_10a_), 130.5 (C_2_), 130.9 (C_6a_), 131.7 (C_16_, C_18_), 132.0 (C_15_, C_19_), 132.1 (C_4a_), 137.4 (C_14_), 140.6 (C_12a_), 164.1 (COO), 183.2 (C_13_). Anal. Calcd. for C_26_H_18_BrNO_3_: C, 66.11; H, 3.84; N, 2.97. Found: C, 66.10; H, 3.83; N, 2.96.

##### 2-(2-oxo-2-(3,4,5-Trimethoxyphenyl)ethyl)isoquinolin-2-ium Bromide **15a**

White solid; 49% yield; mp 270–271 °C; IR ν(cm^−1^): 3019, 1675, 1633, 1581, 1415, 1319, 1161, 1122; ^1^H NMR (500 MHz, DMSO-d_6_) δ_ppm_: 3.81 (s, 3H, OMe), 3.91 (s, 6H, 2 × OMe), 6.67 (s, 2H, H_11_), 7.42 (s, 2H, H_14_, H_18_), 8.13 (t, *J* = 7.0 Hz, 1H, H_6_), 8.34 (t, *J* = 7.5 Hz, 1H, H_5_), 8.43 (d, *J* = 8.5 Hz, 1H, H_4_), 8.56 (d, *J* = 8.5 Hz, 1H, H_7_), 8.70 (d, *J* = 6.5 Hz, 1H, H_3_), 8.74 (d, *J* = 7.0 Hz, 1H, H_2_), 10.03 (s, 1H, H_8_). ^13^C NMR (125 MHz, DMSO-d_6_) δ_ppm_: 56.4 (2 × OMe), 60.4 (OMe), 66.1 (C_11_), 106.1 (C_14_, C_18_), 125.6 (C_3_), 126.9 (C_10_), 127.5 (C_4_), 128.8 (C_13_), 130.7 (C_7_), 131.5 (C_6_), 136.3 (C_2_), 137.3 (C_5_), 137.5 (C_9_), 143.2 (C_16_), 151.7 (C_8_), 153.1 (C_15_, C_17_), 189.9 (C_12_). Anal. Calcd. for C_20_H_20_BrNO_4_: C, 57.43; H, 4.82; N, 3.35. Found: C, 57.42; H, 4.80; N, 3.36.

##### 2-(2-oxo-2-(3,5-Trimethoxyphenyl)ethyl)isoquinolin-2-ium Bromide **15b**

Beige solid; 61% yield; mp 265–268 °C; IR ν(cm^−1^): 3009, 2841, 1685, 1639, 1597, 1463, 1356, 1294, 1208, 1156, 1026; ^1^H NMR (500 MHz, DMSO-d_6_) δ_ppm_: 3.87 (s, 6H, 2 × OMe), 6.66 (s, 2H, H_11_), 6.95 (t, *J* = 2.0 Hz, 1H, H_16_),7.23 (d, *J* = 2.0 Hz, 2H, H_14_, H_18_), 8.12 (t, *J* = 8.0 Hz, 1H, H_6_), 8.34 (t, *J* = 7.5 Hz, 1H, H_5_), 8.42 (d, *J* = 8.0 Hz, 1H, H_4_), 8.55 (d, *J* = 8.0 Hz, 1H, H_7_), 8.70 (d, *J* = 7.0 Hz, 1H, H_3_), 8.75 (d, *J* = 6.5 Hz, 1H, H_2_), 10.05 (s, 1H, H_8_). ^13^C NMR (125 MHz, DMSO-d_6_) δ_ppm_: 55.8 (2 × OMe), 66.3 (C_11_), 106.2 (C_14_, C_18_), 106.3 (C_16_), 125.5 (C_3_), 126.9 (C_10_), 127.4 (C_4_), 130.6 (C_7_), 131.5 (C_6_), 135.5 (C_13_), 136.3 (C_2_), 137.3 (C_5_), 137.5 (C_9_), 151.7 (C_8_), 160.9 (C_15_, C_17_), 190.8 (C_12_). Anal. Calcd. for C_19_H_8_BrNO_3_: C, 58.78; H, 4.67; N, 3.61. Found: C, 58.76; H, 4.67; N, 3.62.

##### 2-(2-oxo-2-(3,4-Trimethoxyphenyl)ethyl)isoquinolin-2-ium Bromide **15c**

Beige solid; 43% yield; mp 259–261 °C; IR ν(cm^−1^): 3039, 3008, 2944, 1675, 1641, 1592, 1510, 1279, 1153, 1011; ^1^H NMR (500 MHz, DMSO-d_6_) δ_ppm_: 3.86 (s, 3H, OMe), 3.92 (s, 3H, OMe), 6.62 (s, 2H, H_11_), 7.25 (d, *J* = 8.5 Hz, 1H, H_17_), 7.55 (d, *J* = 1.5 Hz, 1H, H_14_), 7.82 (dd, *J* = 8.5; 1.5 Hz, 1H, H_18_), 8.12 (t, *J* = 8.0 Hz, 1H, H_6_), 8.33 (t, *J* = 7.5 Hz, 1H, H_5_), 8.42 (d, *J* = 8.5 Hz, 1H, H_4_), 8.55 (d, *J* = 8.5 Hz, 1H, H_7_), 8.68 (d, *J* = 7.0 Hz, 1H, H_3_), 8.75 (d, *J* = 7.0 Hz, 1H, H_2_), 10.03 (s, 1H, H_8_). ^13^C NMR (125 MHz, DMSO-d_6_) δ_ppm_: 55.8 (OMe), 56.0 (OMe), 65.8 (C_11_), 110.3 (C_14_), 111.3 (C_17_), 123.5 (C_18_), 125.5 (C_3_), 126.3 (C_13_), 126.9 (C_10_), 127.4 (C_4_), 130.6 (C_7_), 131.4 (C_6_), 136.4 (C_2_), 137.2 (C_5_), 137.5 (C_9_), 148.9 (C_15_), 151.7 (C_8_), 154.4 (C_16_), 189.2 (C_12_). Anal. Calcd. for C_19_H_8_BrNO_3_: C, 58.78; H, 4.67; N, 3.61. Found: C, 58.77; H, 4.65; N, 3.63.

##### 2-(2-(4-Bromophenyl)-2-oxoethyl)isoquinolin-2-ium **15d**

Physical and spectral data are similar to the ones reported in [6].

##### Ethyl 3-(3,4,5-Trimethoxybenzoyl)pyrrolo[2,1-*a*]isoquinoline-1-carboxylate **16a**

White solid; 56% yield; mp 223–225 °C; IR ν(cm^−1^): 2945, 1707, 1631, 1579, 1456, 1370, 1187, 1134, 761; ^1^H NMR (500 Mz, CDCl_3_) δ_ppm_: 1.41 (t, *J* = 7.0 Hz, 3H, CH_2_CH_3_), 3.93 (s, 6H, 2 × OMe), 3.97 (s, 3H, OMe), 4.41 (q, *J* = 7.0 Hz, 2H, CH_2_CH_3_), 7.15 (s, 2H, H_15_, H_19_), 7.28 (d, *J* = 7.5 Hz, 1H, H_8_), 7.66–7.68 (overlapped signals, 2H, H_7_, H_6_), 7.77 (m, 1H, H_5_), 7.90 (s, 1H, H_2_), 9.57 (d, *J* = 7.5 Hz, 1H, H_9_), 9.84 (dd, *J* = 7.0; 2.0 Hz, 1H, H_4_). ^13^C NMR (125 MHz, CDCl_3_) δ_ppm_: 14.6 (CH_2_CH_3_), 56.5 (2 × OMe), 60.8 (CH_2_CH_3_), 61.2 (OMe), 107.1 (C_15_, C_19_), 110.4 (C_3_), 115.7 (C_8_), 123.3 (C_1_), 124.8 (C_11_), 125.1 (C_9_), 126.9 (C_5_), 128.0 (C_6_), 128.3 (C_4_), 129.5 (C_7_), 129.9 (C_2_), 130.7 (C_12_), 135.0 (C_14_), 137.1 (C_10_), 141.6 (C_17)_, 153.1 (C_16_, C_18_), 164.8 (C_20_), 185.1 (C_13_). Anal. Calcd. for C_25_H_23_NO_6_: C, 69.27; H, 5.35; N, 3.23. Found: C, 69.28; H, 5.33; N, 3.25.

##### Ethyl 3-(3,5-Trimethoxybenzoyl)pyrrolo[2,1-*a*]isoquinoline-1-carboxylate **16b**

Yellow solid; 63% yield; mp 227–229 °C; IR ν(cm^−1^): 2974, 1717, 1631, 1596, 1458, 1370, 1189, 754; ^1^H NMR (500 Mz, CDCl_3_) δ_ppm_: 1.41 (t, *J* = 7.0 Hz, 3H, CH_2_CH_3_), 3.87 (s, 6H, 2 × OMe), 4.41 (q, *J* = 7.0 Hz, 2H, CH_2_CH_3_), 6.70 (t, *J* = 2.5 Hz, 1H, H_17_), 6.99 (d, 2H, *J* = 2.5 Hz, H_15_, H_19_), 7.29 (d, *J* = 7.5 Hz, 1H, H_8_), 7.66–7.68 (overlapped signals, 2H, H_7_, H_6_), 7.78 (m, 1H, H_5_), 7.88 (s, 1H, H_2_), 9.64 (d, *J* = 7.5 Hz, 1H, H_9_), 9.83 (dd, *J* = 7.0; 2.5 Hz, 1H, H_4_). ^13^C NMR (125 MHz, CDCl_3_) δ_ppm_: 14.6 (CH_2_CH_3_), 55.8 (2 × OMe), 60.8 (CH_2_CH_3_), 104.3 (C_17_), 107.3 (C_15_, C_19_), 110.5 (C_3_), 115.7 (C_8_), 123.4 (C_1_), 124.8 (C_11_), 125.3 (C_9_), 126.9 (C_5_), 128.0 (C_4_), 128.3 (C_6_), 129.5 (C_7_), 130.2 (C_2_), 130.7 (C_12_), 137.2 (C_10_), 141.9 (C_14_), 160.8 (C_16_, C_18_), 164.8 (C_20_), 185.7 (C_13_). Anal. Calcd. for C_24_H_21_NO_5_: C, 71.45; H, 5.25; N, 3.47. Found: C, 71.46; H, 5.26; N, 3.48.

##### Ethyl 3-(3,4-Trimethoxybenzoyl)pyrrolo[2,1-*a*]isoquinoline-1-carboxylate **16c**

Yellow solid; 41% yield; mp 219–220 °C; IR ν(cm^−1^): 2979, 1698, 1629, 1598, 1503, 1368, 1190, 750; ^1^H NMR (500 Mz, CDCl_3_) δ_ppm_: 1.41 (t, *J* = 7.0 Hz, 3H, CH_2_CH_3_), 3.98 (s, 3H, OMe), 4.00 (s, 3H, OMe), 4.41 (q, *J* = 7.0 Hz, 2H, CH_2_CH_3_), 6.99 (d, *J* = 8.0 Hz, 1H, H_18_), 7.25 (d, *J* = 7.5 Hz, 1H, H_8_), 7.49 (d, *J* = 2.0 Hz, 1H, H_15_), 7.54 (dd, *J* = 8.5; 2.0 Hz, 1H, H_19_), 7.63–7.68 (overlapped signals, 2H, H_7_, H_6_), 7.76 (dd, *J* = 8.5; 2.0 Hz, 1H, H_5_), 7.85 (s, 1H, H_2_), 9.53 (d, *J* = 7.5 Hz, 1H, H_9_), 9.85 (dd, *J* = 7.5; 1.5 Hz, 1H, H_4_). ^13^C NMR (125 MHz, CDCl_3_) δ_ppm_: 14.6 (CH_2_CH_3_), 56.2 (OMe), 56.3 (OMe), 60.7 (CH_2_CH_3_), 110.1 (C_3_), 110.2 (C_18_), 112.0 (C_15_), 115.5 (C_8_), 123.7 (C_1_), 124.0 (C_19_), 124.9 (C_11_), 125.2 (C_9_), 126.9 (C_5_), 127.9 (C_6_), 128.2 (C_4_), 129.3 (C_7_), 129.4 (C_2_), 130.6 (C_12_), 132.5 (C_14_), 136.9 (C_10_), 149.1 (C_16_), 152.7 (C_17_), 164.9 (C_20_), 184.9 (C_13_). Anal. Calcd. for C_24_H_21_NO_5_: C, 71.45; H, 5.25; N, 3.47. Found: C, 71.43; H, 5.24; N, 3.49.

##### Ethyl 3-(4-Bromobenzoyl)pyrrolo[2,1-*a*]isoquinoline-1-carboxylate **16d**

Yellow solid; 61% yield; ^1^H NMR (500 Mz, CDCl_3_) δ_ppm_: 1.41 (t, *J* = 7.0 Hz, 3H, CH_2_CH_3_), 4.41 (q, *J* = 7.0 Hz, 2H, CH_2_CH_3_), 7.28 (d, *J* = 7.5 Hz, 1H, H_8_), 7.66–7.69 (overlapped signals, 4H, H_16_, H_18_, H_7_, H_6_), 7.73–7.76 (overlapped signals, 3H, H_5_, H_15_, H_19_), 7.77 (s, 1H, H_2_), 9.61 (d, *J* = 8.5 Hz, 1H, H_9_), 9.83 (dd, *J* = 7.0; 2.5 Hz, 1H, H_4_). ^13^C NMR (125 MHz, CDCl_3_) δ_ppm_: 14.6 (CH_2_CH_3_), 60.9 (CH_2_CH_3_), 110.7 (C_3_), 115.9 (C_8_), 123.1 (C_1_), 124.7 (C_11_), 125.1 (C_9_), 126.8 (C_17_), 126.9 (C_5_), 128.0 (C_4_), 128.3 (C_6_), 129.6 (C_7_), 130.0 (C_2_), 130.7 (C_12_), 130.9 (C_15_, C_19_), 131.8 (C_16_, C_18_), 137.3 (C_10_), 138.8 (C_14_), 164.6 (C_20_), 184.8 (C_13_). Anal. Calcd. for C_22_H_16_BrNO_3_: C, 62.57; H, 3.82; N, 3.32. Found: C, 62.55; H, 3.81; N, 3.34.

### 3.2. Biological Activity

#### 3.2.1. Anticancer Activity

Compounds were tested against a panel of 60 human cancer cell lines at the National Cancer Institute, Rockville, MD. Cytotoxicity experiments were performed using a 48 h exposure protocol consisting of a sulforhodamine B assay [41,42,43], described in detail in our previous work [6].

#### 3.2.2. Tubulin Polymerization Assay

A tubulin polymerization assay kit (Cytoskeleton Inc. Denver, CO, USA, Cat. # BK006P) was used to study microtubule assembly according to the manufacturer’s instructions [75,76]. Tubulin polymerization was monitored using a FLUOstar Omega multi-mode microplate reader (BMG LABTECH). The final buffer for tubulin polymerization contained 80 mM PIPES (piperazine-N,N’-bis(2-ethanesulfonic acid)sesquisodium salt) pH = 6.9, 2 mM MgCl_2_, 0.5 mM EGTA (ethylene glycol-bis(β-aminoethyl ether)-N,N,N’,N’-tetraacetic acid), 1 mM GTP, and 10.2% glycerol. Test compounds were added in one single concentration (10 μM) and all compounds except purified tubulin were heated to 37 °C. The reaction was initiated by the addition of tubulin to a final concentration of 3.0 mg/mL. Paclitaxel and phenstatin were used as positive controls under the same conditions. Absorbance was measured at 340 nm for 1 h at 1 min intervals at 37 °C.

### 3.3. Molecular Modeling

#### 3.3.1. Molecular Docking

Flexible-ligand docking experiments were performed in two steps. First, blind docking was performed using Autodock 4.2 [77] using a 90 × 118 × 114 gridbox with 0.825 Å spacing (total search volume of 998,811 Å^3^, as to include the entire heterodimer, PDB ID: 4O2B [63]) and was centered on the entire structure (x = 18.997, y = 68.705, z = 46.491). Co-crystallized GTP, GDP, Ca^2+^, and Mg^2+^ ions were kept during receptor preparation, and the target protein was kept rigid during all docking experiments. The 3D structures of colchicine, phenstatin, and compound **10a** were constructed in Avogadro v1.2.0 [78] and were energetically optimized in the MMFF94 force field until a local energy minimum was reached. A total of 8 jobs of 2000 runs were performed for each of the 3 ligands using the Lamarckian Genetic Algorithm (LGA) for a total of 16,000 runs per ligand. Conformations having theoretical binding energies lower than −5.5 kcal/mol were clustered and ranked using an RMSD tolerance of 2.0 Å in order to select the lowest-binding representatives for further binding pose refinement. Cluster representatives with binding energies lower than −7.0 kcal/mol were visually inspected for binding orientation.

A smaller gridbox centered on the lowest binding cluster of compound **10a** from blind docking (58 × 58 × 58 points, 0.375 Å spacing, x = 14.316, y = 67.184, z = 44.464) was chosen for local docking. In this case, 1,000 poses were generated per ligand in a single run, and results were clustered and ranked based on theoretical binding energy. RMSD between re-docked and co-crystallized colchicine ligand was used as quality control for the docking protocol. Visual inspection and molecular graphics were made in the PyMOL Molecular Graphics System, Version 2.5.0. (Schrödinger, LLC, New York, NY, USA). Furthermore, 2D ligand interaction diagrams were generated in Maestro, release 2018-4 (Schrödinger, LLC, New York, NY, USA).

#### 3.3.2. Molecular Dynamics Simulations

Each of the three lowest-scoring cluster representatives of compound **10a** obtained from local docking experiments was subjected to MD simulations in the colchicine binding site of the α,β-tubulin heterodimer (chains A and B of PDB ID: 4O2B). The BioLuminate^®^ (Schrödinger, Inc., New York, NY, USA) graphical interface [79] was used for initial system construction, while the subsequent MD simulations were performed with the Desmond (D.E. Shaw Research) software [80]. The missing loop of tubulin chain B (residues 276–281) was modeled using MODELLER [81], as implemented in UCSF Chimera release 1.15 [82]. Proteins, ligands, and ions were described using the OPLS-AA force field [83,84]. The **10a**/tubulin complexes were solvated using a TIP3P water model [85] in orthorhombic simulation boxes while allowing for a minimum distance of 10 Å between the complex and the walls of the simulation box. Ions were added to electrostatically neutralize the systems and to attain a salt concentration of 0.15 M NaCl. After initial geometry optimization, the standard Desmond solute relaxation protocol was applied. This protocol involves sequential short MD simulations using restraints of decreasing magnitude on atomic positions, followed by short restraint-free equilibration simulations. In the production runs, each complex was simulated for 10 ns, and system configurations were saved every 5 ps for analysis. Throughout the simulations, the NPT ensemble (i.e., constant number of particles, pressure, and temperature) was employed, using the Nosé–Hoover chain thermostat (300 K) [86] and Martyna–Tobias–Klein barostat (1 atm) [87]. The MD trajectories were analyzed with respect to protein backbone and ligand RMSD, as well as protein-ligand contacts. Protein-ligand interaction fraction plots were generated in R using the ggplot2 package [88]. Configurational entropy variations were evaluated from 25 ns long MD simulations of the free compound in solution and bound to tubulin, respectively. Schlitter’s formula was used for the configurational entropy calculation [89]. Prior to entropy calculations using the GROMACS 2020.3 modelling suite [90], all the structures in the MD trajectories were geometrically fitted to the first frame in order to remove the rotational and translational degrees of freedom. The fitting procedure included all heavy atoms of compound **10a**. To evaluate the convergence of the entropy profile, the calculations included steeply increasing time intervals along the entire MD trajectories.

#### 3.3.3. In Silico ADME and Toxicity Predictions

The ADME in silico evaluation for the most active compound **10a** was performed using the Swiss ADME web tool (http://swissadme.ch/index.php (accessed on 29 March 2023)) in terms of molecular properties, pharmacokinetics, drug-likeness, and medicinal chemistry.

The in silico toxicological evaluation for the most active compound **10a** was performed using the web-service Cell-Line Cytotoxicity Predictor (CLC-Pred), which screens for in silico cytotoxicity on a panel of 278 tumor cells and 27 normal human cell lines from different tissues [91].

## 4. Conclusions

Five series of potential microtubule-targeting anticancer pyrrolo-fused heterocycles were designed as analogs of phenstatin and synthesized through 1,3-dipolar cycloaddition. Their antiproliferative activity was tested against NCI’s panel of 60 cancer cell lines. Pyrrolo[1,2-*a*]quinoline **10a** proved to be the best in terms of growth inhibition properties, with almost all GI_50_ values in the submicromolar range. The best anticancer behavior was displayed on the A498 renal cell line (GI_50_ = 27 nM), melanoma MDA-MB-435 cell line (GI_50_ = 35 nM), and non-small cell lung cancer NCI-H522 line (GI_50_ = 50 nM). In vitro, tubulin polymerization inhibitory properties were found for compound **10a**, while PRISM analysis indicated a strong profile correlation to other microtubule inhibitors (even if with a totally different structure). QSAR simulation of ADMET properties showed a promising drug-likeness profile for compound **10a** with very good selectivity for tumor cells, good physicochemical properties, good GI absorption, moderate solubility, and adequate bioavailability. The binding mode of **10a** was investigated through molecular docking and molecular dynamics simulations, as well as configurational entropy calculations. This compound was confirmed to bind to the colchicine site of tubulin through global and local docking experiments, but molecular dynamics simulations revealed that some of the predicted interactions in the docking experiments were not stable. As such, the lowest-scoring conformation from local docking, BM I, drifted away from its original orientation during MD simulations, while BM III, which was also identified through global docking experiments, was the most stable throughout the MD simulation. The second-lowest scoring conformation, BM II, which was the most similar in terms of relative orientation and hydrophobic amino acid interactions to other co-crystallized trimethoxy-substituted ring-containing tubulin binders including colchicine, was also stable throughout the MD simulation. However, longer MD simulations are necessary in order to demonstrate the full stability of the investigated systems. Configurational entropy calculations indicated similar values for configurational entropy loss upon binding for all three docked solutions, suggesting that all three binding modes are equally favorable in terms of entropic contribution. Taken together, our results suggest that for compound **10a**, docking experiments alone are not sufficient for the adequate description of molecular interaction details in terms of target binding, which makes subsequent scaffold optimization more difficult to achieve. This highlights that a thorough in silico investigation should be performed for novel pyrrole-fused agents in order to have more meaningful insights into the molecular details of anticancer activity. Hopefully, our efforts will guide the discovery of novel potent antiproliferative compounds with pyrrolo-fused heterocyclic cores, especially from an in silico perspective.

## Data Availability

Data is contained within the article and Appendix A.

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
