# Peer review of "Exploring Pyrrolo-Fused Heterocycles as Promising Anticancer Agents: An Integrated Synthetic, Biological, and Computational Approach"

_pharmaceuticals, 2023, doi:10.3390/ph16060865_

Round 1

Reviewer 1 Report

It is not very clear how the authors confirm regioselectivity of cycloaddition between pyrimidinium salts and acetylenes where two possible isomers can be formed. Nothing is given in experimental part. Without mentioning pf this it is difficult to accept this paper

It is not very clear how the authors confirm regioselectivity of cycloaddition between pyrimidinium salts and acetylenes where two possible isomers can be formed. Nothing is given in experimental part. Without mentioning pf this it is difficult to accept this paper

Author Response

First of all, we want to thank you for the availability to thoroughly inspect our manuscript. We highly appreciate your valuable suggestions.

As researchers who have worked extensively in the field of cycloaddition of ylides, we believe that our previous work in the field of cycloaddition of ylides, which did provide more extensive experimental details, serves as evidence of our expertise and supports the validity of our current findings. However, as this manuscript is primarily focused on the medicinal applications of the cycloaddition reaction, we chose to emphasize the biological and pharmacological aspects of our work. Our previous articles in this field have provided evidence through NMR, IR, and RDX, but also computational studies that support the formation of only the regioisomer also presented in this manuscript, when using non-symmetric substituted acetylenes (http://dx.doi.org/10.1016/j.bmc.2015.03.077; http://dx.doi.org/10.1016/j.ejmech.2013.09.061; https://doi.org/10.1016/j.ejmech.2013.04.069; http://dx.doi.org/10.3109/14756366.2015.1039530; 10.1002/jhet.5570400213; 10.1002/jhet.5570410621).

However, we have included information in the manuscript about the selective formation of one regioisomer, which is in agreement with the electronic effects within ethyl propiolate and ylide species (in yellow in sections 2.1 and 3.1.2)

Reviewer 2 Report

The manuscript by Danac and co-workers submitted for publication in “Pharmaceuticals” describes five series of phenstatin analogs. These potential microtubule-targeting anticancer pyrrolo-fused heterocycles were synthesized by using 1,3-dipolar cycloaddition as the key synthetic step. Moreover,, the obtained compounds were evaluated for anticancer activity and ability to inhibit tubulin polymerization in vitro. 

Revision Request:

In scheme 5 please indicate R2 in the legend (letter a).

In all the schemes: Please use the same format to indicate the substituents (R1 superscript or R1 subscript)

SI: the acronym RMN is uncorrect; please use the correct acronym for NMR.

Although the quality of English language is not bad, moderate editing is required.

Author Response

First of all, we want to thank you for the availability to thoroughly inspect our manuscript. We highly appreciate your valuable suggestions. 

Revision Request:

In scheme 5 please indicate R2 in the legend (letter a).

Thank you for the observation. We corrected it.

In all the schemes: Please use the same format to indicate the substituents (R1 superscript or R1 subscript)

Thank you for the observation. We corrected it.

SI: the acronym RMN is uncorrect; please use the correct acronym for NMR.

Thank you for the observation. We corrected it.

Although the quality of English language is not bad, moderate editing is required.

We revised the manuscript.

Reviewer 3 Report

This manuscript from Amărandi and co-workers report the synthesis and anticancer activity evaluation of five series of pyrrolo-fused heterocycles analogs of microtubule inhibitor Phenstatin, and further investigate the molecular details of the most active compound 10a interaction with tubulin through molecular docking and molecular dynamics simulations, as well as configurational entropy calculations. This work will guide the discovery of novel potent antiproliferative compounds with pyrrolo-fused heterocyclic cores.

The manuscript is generally well written, and the graphical presentation is excellent. The supporting information is also in good shape, which fully supports the results. I consider this work attracts sufficient readership for pharmaceuticals and thus recommend it for publication, provide that the following points are reflected in the revision process.

1.      All “RMN spectrum” in SI should be “NMR spectrum”.

Author Response

First of all, we want to thank you for the availability to thoroughly inspect our manuscript. We highly appreciate your valuable suggestions. 

  1. All “RMN spectrum” in SI should be “NMR spectrum”.

Thank you very much, we corrected it in both, manuscript and supplementary material.

Reviewer 4 Report

In the manuscript, five new series of pyrrole-fused heterocycles were designed through a scaffold hybridization strategy as analogs of the well-known microtubule inhibitor Phenstatin. Compounds were synthesized and some were evaluated for anticancer activity. Not enough in vitro molecular level-based studies were performed. The IC50 of all synthesized compounds over the target protein should be presented and integrated into the manuscript. Then, the most promising compound should be further evaluated. 

No comment on the English

Author Response

First of all, we want to thank you for the availability to thoroughly inspect our manuscript. We highly appreciate your valuable suggestions. 

Not enough in vitro molecular level-based studies were performed. The IC50 of all synthesized compounds over the target protein should be presented and integrated into the manuscript. Then, the most promising compound should be further evaluated.

Obtaining IC50 values over the tubulin for all compounds depends on the availability of resources, such as reagents, and equipment, which may vary depending on the laboratory and institution, and unfortunately for us, it is impossible to obtain in a reasonable time. We used the instruction provided by the kit (Cat. # BK006P) purchased from Cytoskeleton Inc., to determine the ability of the most promising compound 10a to interact with tubulin.

Given these constraints, we believe that the data we provided in our submission are still relevant and valuable for the scientific community. They demonstrate the potential of our compounds to inhibit the growth of human cancer cells and provide some insights into their structure-activity relationship.

Round 2

Reviewer 1 Report

It is still not proved in THIS paper the direction of cycloadiotion between pyrimidinium ylides and acetylene (2 or 6 of pyrimidine ring)

Reviewer 4 Report

Since the authors were not able to perform the requested experiemtents due to lack of facility and the limited revision timeframe, the manuscript can not be published in the current form in this journal.

No major comments related to the English editing.